# Recent Advances in the Tunable Optoelectromagnetic Properties of PEDOTs

**DOI:** 10.3390/molecules30010179

**Published:** 2025-01-04

**Authors:** Ling Zhu, Qi Liu, Yuqian Zhang, Hui Sun, Shuai Chen, Lishan Liang, Siying An, Xiaomei Yang, Ling Zang

**Affiliations:** 1School of Pharmacy and Flexible Electronics Innovation Institute, Jiangxi Science & Technology Normal University, Nanchang 330013, China; zhuling0620@163.com (L.Z.); liuqi1903794147@163.com (Q.L.); ansiying2023@163.com (S.A.); 2Jiangxi Provincial Key Laboratory of Flexible Electronics, Nanchang 330013, China; 13766287190@163.com (Y.Z.); lianglishan09@163.com (L.L.); 3Binzhou Testing Center, Binzhou 256600, China; shlbl1990@126.com; 4Department of Materials Science and Engineering, University of Utah, Salt Lake City, UT 84112, USA; jaimee@eng.utah.edu

**Keywords:** conducting polymer, PEDOT, PEDOT:PSS, photoelectronic, magnetic, optoelectromagnetic

## Abstract

Conducting polymers represent a crucial class of functional materials with widespread applications in diverse fields. Among these, poly(3,4-ethylenedioxythiophene) (PEDOT) and its derivatives have garnered significant attention due to their distinctive optical, electronic, and magnetic properties, as well as their exceptional tunability. These properties often exhibit intricate interdependencies, manifesting as synergistic, concomitant, or antagonistic relationships. In optics, PEDOTs are renowned for their high transparency and unique photoelectric responses. From an electrical perspective, they display exceptional conductivity, thermoelectric, and piezoelectric performance, along with notable electrochemical activity and stability, enabling a wide array of electronic applications. In terms of magnetic properties, PEDOTs demonstrate outstanding electromagnetic shielding efficiency and microwave absorption capabilities. Moreover, these properties can be precisely tailored through molecular structure modifications, chemical doping, and composite formation to suit various application requirements. This review systematically examines the mechanisms underlying the optoelectromagnetic properties of PEDOTs, highlights their tunability, and outlines prospective research directions. By providing critical theoretical insights and technical references, this review aims to advance the application landscape of PEDOTs.

## 1. Introduction

Conducting polymers (CPs), also known as intrinsically conductive polymers, have transformed the understanding of organic polymers by redefining them from insulating “plastics” to functional conductive materials [1]. The discovery and development of doped polyacetylene, with its high electrical conductivity but poor stability, initiated extensive efforts to advance CPs with improved performance, structural stability, and scalability. Prominent examples of CPs include polyaniline (PANi), polypyrrole (PPy), polythiophene (PTh), and their derivatives (Figure 1a) [2]. Among these, poly(3,4-ethylenedioxythiophene) (PEDOT, Figure 1b), first synthesized in 1988 by Bayer AG, marked a significant breakthrough. PEDOT addressed limitations of PTh, such as low conductivity and poor oxidative stability, by combining high conductivity, optical transparency, and structural stability [3]. With a small energy gap and high work function, PEDOT has found widespread applications in organic electronics, including capacitors [4,5], organic solar cells [6], electrochromic films [7], and electromagnetic shielding coatings [8]. However, its insolubility and non-melting nature have posed challenges for large-scale processing and applications.

To overcome these issues, poly(3,4-ethylenedioxythiophene):polystyrene sulfonate (PEDOT:PSS, Figure 1c) was developed by incorporating polystyrene sulfonate (PSS) during the chemically oxidative polymerization of EDOT monomers. PEDOT:PSS is commercially produced on a large scale, forming stable water-dispersible suspensions (Figure 1c) with excellent film-forming properties, high conductivity, and optical transparency. Initially used by Agfa for antistatic coatings in photographic films [9], PEDOT:PSS exhibits unique properties, including mixed electronic and ionic conductivity, adjustable work function, low magnetization, high mechanical flexibility, and superior thermal and electrochemical stability [10,11]. Furthermore, it demonstrates remarkable biocompatibility and adaptability to various processing techniques—such as physical coating [12], spinning [13], and inkjet printing [14]—enabling the creation of films [15], hydrogels [16], aerogels [17], fibers [18], powders [19], and elastomers [20]. These characteristics have positioned PEDOT:PSS as one of the most widely used CPs in organic electronics, garnering significant academic and industrial interest.

**Figure 1 molecules-30-00179-f001:**
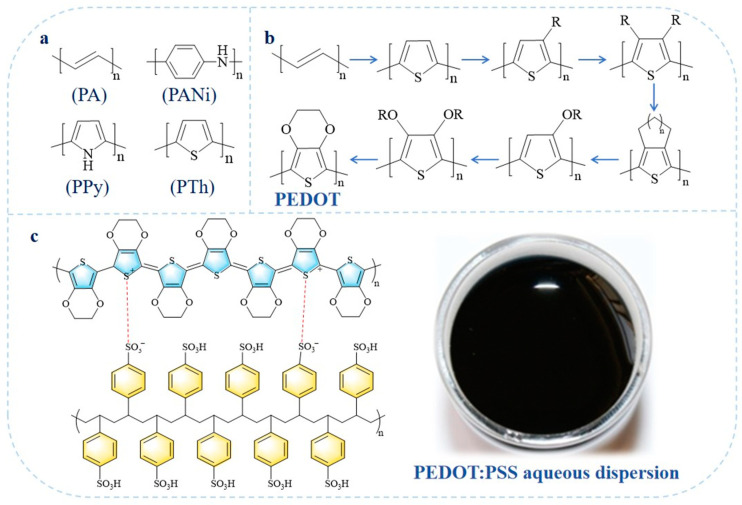
(**a**) Molecular structures of common CPs, (**b**) structure evolution from polyacetylene to PEDOT, and (**c**) molecular structure of PEDOT:PSS along with a photograph of its aqueous dispersion.

Beyond PEDOT:PSS, advances in PEDOT derivatives, achieved through side-group substitution or main-chain modifications, have further enhanced solution processability and broadened functionality [21,22,23]. Over the past decade, numerous studies have explored CPs, particularly PEDOT and its derivatives (collectively referred to as PEDOTs), in applications ranging from traditional uses like antistatic coatings [24] and solid capacitors [25] to emerging technologies such as organic light-emitting diodes (OLEDs) [26,27], organic thin-film transistors (OTFTs) [28,29], electrochromic devices (ECDs) [30,31,32], organic solar cells (OSCs) [33,34], thermoelectric devices [35,36,37], solar water purification systems [38,39], supercapacitors [40,41,42], printed circuit boards (PCBs) [43], anticorrosive coatings [44], electromagnetic shielding (EMS) coatings [45,46,47], and various sensor technologies [48,49].

The exceptional and tunable optoelectromagnetic properties (optical, electrical, and magnetic) of PEDOTs underpin their success, serving as a foundation for material and device design. These properties are intrinsically tied to the physicochemical and mechanical characteristics of PEDOTs [50], and are often interdependent, exhibiting synergistic or antagonistic relationships depending on application demands. Significant research has focused on optimizing PEDOTs’ molecular structure, synthesis methods, doping strategies, composites, and surface modifications to enhance their key properties [51,52,53]. Despite these advances, challenges persist in precisely regulating photoelectric properties while balancing mechanical performance.

Previous reviews have predominantly concentrated on optimizing either the electrical or mechanical properties of PEDOTs, often emphasizing PEDOT:PSS. However, in the past five years, research interest has shifted towards exploring PEDOTs’ applications in rapidly evolving fields such as energy electronics [54], flexible electronics [55], wearable electronics [56], and bioelectronics [57]. Surprisingly, despite rapid progress, no comprehensive review has addressed the optoelectromagnetic properties of PEDOTs and their intrinsic relationships with fundamental material characteristics. This review aims to fill this gap by providing a comprehensive analysis of recent advancements in regulating PEDOTs’ optoelectromagnetic properties to meet diverse application requirements. It seeks to uncover the intrinsic relationships between these properties and the fundamental nature of PEDOTs, offering insights into their future development. Additionally, this review identifies current challenges and proposes potential directions for innovation in this dynamic field.

## 2. Optical Properties and Structural Tunability of PEDOTs

PEDOTs are highly efficient in absorbing photons and generating charge carriers, with the majority of their applications utilizing them in film form. Their optical properties encompass absorbance, transmittance, reflectivity, scattering, and refractive index—all of which are critical for their role in optoelectronic devices [58,59,60]. These properties are rarely utilized in isolation but are typically integrated with their conductive properties, resulting in synergistic effects that enhance device performance. Applications leveraging these combined properties include touch screens, transparent electrodes, OLEDs, OTFTs, ECDs, OSCs, and solar water purification systems (SWP) [61,62].

### 2.1. Spectral Absorption Property

The spectral absorption behavior of PEDOTs in the ultraviolet–visible (UV–vis) region is primarily reflected in their coloration and changes in film transmittance. PEDOT efficiently absorbs photons across a broad spectral range (400–1690 nm) [58]. Unlike dyes or pigments, PEDOT’s oxidation and reduction states are closely linked to its doping and dedoping processes, which directly impact both its color and electrical conductivity. In the oxidized state, portions or the entirety of the thiophene rings in the molecular chain are oxidized, generating positive charges along the chain. This state results in a more uniform charge distribution, reduced light absorption, and higher conductivity, giving PEDOT a lighter color, such as transparent or light blue. Conversely, in the reduced state, the thiophene rings lose their positive charges, leading to increased light absorption, reduced conductivity, and a darker color, such as dark blue [59,62].

The doping process introduces counterions (e.g., PSSNa or other acidic salts) into the PEDOT molecular chains, facilitating the oxidation of thiophene rings. During dedoping, the electronic interactions between the dopant and the PEDOT chains weaken or vanish, returning the charge distribution to its original state [63]. The doping level significantly influences PEDOT’s coloration. For instance, as the PSS content in PEDOT:PSS increases, its color deepens progressively from light to dark [64]. In specific cases, depending on the preparation method and dopants used, PEDOT may even appear black—for example, in Orgacon^®^ EL-P 4020 (Agfa-Gevaert N.V., Belgium), a screen-printable conductive ink. Beyond chemical processes, electrical doping and dedoping can also modulate PEDOT’s coloration, accompanied by changes in its UV–vis absorption spectrum. In electrochromic applications, for example, precise control over PEDOT’s color can be achieved by adjusting the operating voltage or current [65].

A widely adopted strategy for regulating the color of PEDOT-based materials involves modifications to the polymer backbone. This approach encompasses the design and synthesis of a diverse range of PEDOT derivatives, analogs, copolymers, and novel structures. For instance, poly(2-aminomethyl-3,4-ethylenedioxythiophene) (PEDOT-MeNH_2_) (Figure 2a) demonstrates a significant color shift, transitioning from oxidized violet to blue. This derivative exhibits excellent contrast (41.8%) and coloring efficiency (152.1 cm^2^/C) [66]. Extensive research has focused on designing PEDOT analogs by altering the ring structure, substituting one or both oxygen atoms in the peripheral ring with heteroatoms (e.g., sulfur or nitrogen), or replacing the sulfur in the thiophene core. Examples of these analogs include poly(3,4-propylenedioxythiophene) (PProDOT), poly(thieno[3,4-b]-1,4-oxathiane) (PEOTT), poly(3,4-ethylenedithiathiophene) (PEDTT), poly(N-methyl-3,4-dihydrothieno[3,4-b][1,4]oxazine) (PMDTO), and poly(N-ethyl-3,4-dihydrothieno[3,4-b][1,4]oxazine) (PEDTO) (Figure 2a) [67].

Copolymerization strategies, such as combining 3,4-propylenedioxythiophene (ProDOT) with 3,4-ethylenedioxythiophene (EDOT), have yielded solution-processable polymers with highly tunable optical and electronic properties. Among these, ProDOT-EDOT_2_ (Figure 2a) has garnered attention for its unique properties, combining the low oxidation potential, redox behavior, and deep-blue neutral color of PEDOT with the high solubility and exceptional electrochromic properties of PProDOT due to alkoxyl functionalization [68]. Additionally, PProDOT is a versatile material for biosensing applications, attributed to its excellent biocompatibility, non-toxicity, and biodegradability [69]. Additionally, hybrid poly(selenophene-EDOT) (Figure 2a) materials exhibit promising electrochromic properties, benefiting from the lower bandgap and improved planarity of poly(selenophene) combined with the high conductivity, transparency, and stability of PEDOT [67].

These PEDOT derivatives, analogs, and copolymers showcase significant advancements in electrochromic properties, achieved through careful molecular design and synthesis. Beyond enhanced electrochromic performance, they hold immense potential for various optoelectronic device applications. The materials exhibit excellent thermal and electrochemical stability, facilitating the meticulous modulation of electronic, optical, and redox attributes via structural advancements. These features position PEDOT-based materials to meet the competitive demands of future commercial applications in both electrochromic materials and optoelectronic devices [70,71,72].

Beyond the UV–vis spectrum, PEDOTs also exhibit absorption in the infrared (IR) spectrum, which holds significant research value. For instance, reversible modulation of the infrared spectrum of PEDOT:PSS films can be achieved by treatment with a redox solvent, correlating with changes in its oxidation and reduction states [73]. This property enables PEDOTs to have potential applications in infrared optical security printing, anti-counterfeiting, display technologies, and related fields. For example, Zhang et al. developed an intelligent infrared-rewritable polymer film based on PEDOT:PSS, which supports direct writing and erasing technology in the infrared region (Figure 2b) [51].

The integration of PEDOTs with other materials—such as graphene oxide (GO), vanadium oxide (V_2_O_5_), or titanium dioxide (TiO_2_)—can further modify their spectroscopic absorption properties, expanding their utility in optoelectronics [74,75,76]. For example, PEDOT-based colorimetric chemosensors, which exhibit altered spectral characteristics, can be used to detect specific gasses such as hydrogen peroxide (H_2_O_2_) [77]. These sensors offer advantages including high sensitivity, fast response times, and excellent stability, and they can be combined with chemosensitive mechanisms. Such capabilities make them promising for applications in visual environmental monitoring, safety protection, and related fields [78].

An emerging application involves utilizing PEDOT:PSS-based hydrogels as photothermal conversion materials in solar water purification (SWP) systems. These composite hydrogels, which feature exceptional light absorption and water transport properties alongside high photothermal conversion efficiency, enable efficient water evaporation and purification. This has broad potential for addressing challenges in sewage treatment, seawater desalination, and other water purification processes. For example, Zhao et al. developed a PEDOT:PSS–PVA hydrogel with remarkable properties, including 99.7% light absorption over a broad wavelength range (250–2500 nm), an impressive energy efficiency of 98.0%, and a rapid evaporation rate of 2.84 kg m^−2^ h^−1^ under single solar irradiation [79].

### 2.2. Spectral Transparency Property

In many organic optoelectronic applications, the colors of PEDOTs may be less critical, as their conductive properties are often the primary focus. However, the optical transmittance of PEDOTs is a key factor in thin-film applications, particularly for antistatic films, transparent electrodes, and similar uses [80]. As discussed earlier, doped PEDOT films typically exhibit high transmittance, allowing substantial light passage, particularly in the visible spectrum. For instance, a PET substrate coated with PEDOT achieves a transmittance of 90.8% at 550 nm [81]. Additionally, PEDOT:PSS, after appropriate processing, can be used as a transparent electrode material, making it a potential indium tin oxide (ITO) conductive glass alternative [61].

Achieving simultaneous high transparency and conductivity in PEDOT:PSS films is, however, a significant challenge. A more reliable approach involves combining PEDOT:PSS with highly conductive, light-colored inorganic materials. For example, Bai et al. [82] developed a composite material consisting of silver nanowires, MXene (Ti_3_C_2_T_x_), and PEDOT:PSS (AgNW-MXene@PEDOT:PSS), achieving a film transmittance of 97.6% at 550 nm and a sheet resistance of 17 Ω sq^−1^ (Figure 2c). Additionally, factors such as film thickness, preparation techniques, doping levels, and substrate materials significantly influence the specific transmittance values [83,84].

In photovoltaic devices such as OSCs, high transmittance allows for increased light absorption, which enhances energy conversion efficiency. For instance, Lee et al. [85] prepared hybrid inks of Ag nanowires and PEDOT:PSS in ratios of 1:10, 1:20, and 1:30, which were then bar-coated onto polyurethane substrates. By optimizing the ink ratios, they achieved a film transmittance of up to 88.6% while maintaining excellent mechanical properties, including bending, folding, curling, twisting, and stretching. In transparent conductive layers, high transmittance enhances visual clarity and reduces visual interference [86,87,88]. In ECDs, PEDOT-based films are commonly employed as active electrochromic layers. Their distinct optical transparency and pronounced color changes before and after doping enable high optical contrast, further expanding their utility in optoelectronic applications.

### 2.3. Spectral Reflection Property

Reflectivity is a measure of a material’s surface ability to reflect light. For PEDOT films, reflectivity is influenced by several factors, including surface roughness, film thickness, refractive index, as well as the wavelength and angle of incident light [89,90,91]. A smoother surface typically provides more uniform reflection and lower reflectivity, reducing light scattering and reflection losses, which can enhance the efficiency of optoelectronic devices. Conversely, rough surfaces may increase light scattering and lead to higher reflectivity. Reducing reflectivity is advantageous for improving optoelectronic device performance, as it allows more light to be absorbed or transmitted.

For instance, Futsch et al. [62] developed a PEDOT:PSS/V_2_O_5_ hybrid film exhibiting reversible behavior in a lithium-based electrolyte. This hybrid material showed significant reflection modulation (ΔR of approximately 20.5% at 550 nm) and excellent contrast (ΔE* > 30), with optimal performance achieved at a film thickness of 3.7 μm (Figure 2d). Similarly, Mingzhi et al. [92] deposited a graphene oxide (GO) layer via solution processing between a Ag electrode and a PEDOT:PSS layer in Si/PEDOT:PSS hybrid heterojunction solar cells (HHSCs). The GO layer acted as an anti-reflection coating due to its well-matched refractive index. This configuration resulted in a planar Si/PEDOT:PSS/GO HHSC with the highest power conversion efficiency (PCE) of 13.76%.

Although PEDOT itself is not a light-emitting material, its small energy gap and excellent carrier transport properties make it valuable in optoelectronic light-emitting devices such as quantum dot light-emitting diodes (QD-LEDs) [93] and OLEDs [94]. PEDOT is often employed as a hole injection layer (HIL) or hole transport layer (HTL) material to enhance electroluminescence performance and long-term operational stability [95]. This stability is attributed to the highly stable oxidized state of PEDOT, which maintains its conductivity even after prolonged exposure to 120 °C for 1000 h.

The strong acidity of PEDOT:PSS, resulting from its high sulfonate content, combined with the distribution of hydrophilic PSS on the surface and exposure to air, leads to water absorption, swelling, poor water resistance, and significant hygroscopicity. These drawbacks severely impact the efficiency and stability of organic solar cells, organic light-emitting diodes, and other devices, particularly those with multilayer structures [44]. For instance, the acidity of PSS, derived from its sulfonic acid groups, can trigger electrochemical reactions that result in metal corrosion when in contact with metals like aluminum. Furthermore, the hygroscopicity of PEDOT:PSS facilitates water absorption from the environment, forming a hydration layer that accelerates hydrogen ion migration and diffusion, exacerbating metal corrosion. This damages the performance of metal electrodes, reduces charge collection and injection efficiency, and shortens the device’s lifespan in applications such as OLEDs and OSCs [96].

Additionally, the acidic ions in the hydration layer trigger hydrolysis reactions, which, combined with absorbed water, severely degrade organic materials. This negative synergistic effect destabilizes interfaces within device layers, leading to weakened adhesion, increased delamination, higher interfacial resistance, and impeded charge transfer. Introducing a barrier layer into multilayer device structures is an effective protective strategy. For example, barrier films combining inorganic oxides (e.g., alumina, silicon oxide) with organic polymers (e.g., polyethylene terephthalate, polyimide) leverage the high barrier properties of inorganic materials and the flexibility of organic materials. By optimizing barrier layer preparation processes, such as atomic layer deposition or physical vapor deposition, the thickness, uniformity, and density of the barrier layer can be precisely controlled, effectively blocking water and acid penetration to protect the device’s internal structure.

Another viable approach is the chemical modification of PEDOT:PSS. Acidity can be controlled through chemical doping, where amino-containing additives interact with the sulfonic acid groups in PSS chains, reducing acidity [97,98,99]. Surface modifications of PEDOT:PSS can also reduce hygroscopicity and acidic activity. For example, plasma treatments can introduce hydrophobic groups (e.g., fluorocarbon groups) on the surface to minimize moisture adsorption [100]. Alternatively, chemical grafting reactions can attach basic or neutral functional groups to neutralize acidic sites, mitigating acidic hazards [101,102].

Flexible OLEDs have emerged as a focal point for new-generation displays, owing to their unique characteristics such as flexibility, thinness, self-luminescence, low energy consumption, and superior optical display properties [103]. Acting as a HIL in OLEDs, the PEDOT:PSS film can effectively lower the energy barrier between their anode (ITO, FTO, etc.) and the organic light-emitting layer, facilitating easier hole injection from the former into the latter and thus significantly enhancing the charge injection efficiency [27]. PEDOT:PSS also exhibits excellent charge transport properties, providing an efficient pathway for injected holes traveling through the organic layer. Such rapid charge transport minimizes the accumulation and loss of charges during their transmission, ensuring stable current flow, which ultimately improves the luminous efficiency and stability of OLEDs [104]. Moreover, PEDOT:PSS may further promote the diffusion of holes within the organic layer, enabling a more uniform distribution of holes in the emissive layer. This uniformity enhances the overall utilization of the emissive layer, reduces localized overheating and uneven light emission, and consequently improves the light emission quality and uniformity of OLEDs.

On the other hand, through doping or modification techniques, the energy level structures of PEDOT:PSS can be adjusted to better match that of adjacent organic layers [105,106]. This alignment can optimize the charge injection and transport processes, minimize energy losses, and enhance the efficiency and performance of OLEDs. In flexible OLEDs, the unique advantages of mechanical flexibility and stability of PEDOT:PSS allow the devices to withstand bending, stretching, and other deformations. Therefore, it can effectively mitigate the impact of external stresses on the device, to protect the internal light-emitting layers from damage and ensure the normal operation of flexible OLEDs under various deformation conditions [107,108]. Additionally, PEDOT-based composite films have demonstrated infrared absorption and reflection capabilities under electrical stimulation, which could be harnessed for advanced applications such as information storage [109], encryption [110], and anti-counterfeiting technologies [111].

### 2.4. Spectral Refraction Property

The refractive index describes the relationship between the change in the velocity of light as it propagates through a medium and the angle of incidence. For PEDOT films, the refractive index is influenced by factors such as molecular structure, doping level, and preparation techniques [112,113,114]. Lara et al. investigated the complex refractive indices of seven PEDOT:PSS-based samples, finding that three exhibited isotropic behavior while four demonstrated optical anisotropy [112]. The refractive index (n) values for all samples ranged from 0.8 to 1.65, decreasing with increasing wavelength and exhibiting normal dispersion behavior. For isotropic films, both the average refractive index (n) and the average extinction coefficient (k) increased with film thickness. These optical properties were found to be relatively stable, depending on the film’s composition, structure, and doping state.

For example, Singh et al. modified the optical properties of PEDOT:PSS by incorporating deionized (DI) water, ethylene glycol (EG), and multi-walled carbon nanotubes (MWCNTs) [114]. Upon doping, an increase in the refractive index was observed, suggesting the formation of localized energy states within the energy bandgap. These states act as recombination centers, enhancing low-energy electron transitions. The original PEDOT:PSS film exhibited a refractive index of approximately 1.6, while samples modified with EG showed an increased refractive index of around 1.8. A similar increase was observed in samples doped with MWCNTs (Figure 2e).

Generally, the refractive index decreases with increasing wavelength, reflecting normal dispersion behavior. However, beyond a certain wavelength (above 1000 nm), it begins to increase with wavelength. The refractive index of PEDOT:PSS films can also vary slightly depending on testing conditions and doping strategies, leading to more complex optical behaviors. The magnitude of the refractive index is critical for the application of PEDOT films in optical devices, as it directly affects the propagation path and focusing properties of light.

**Figure 2 molecules-30-00179-f002:**
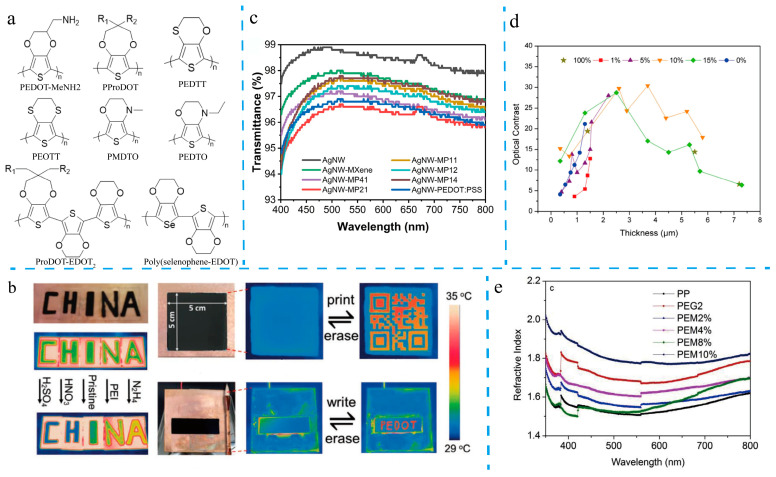
(**a**) Molecular structure of PEDOT derivatives and analogs. (**b**) Direct write and erase infrared images of PEDOT:PSS films [51]. (**c**) UV–vis spectra of AgNW, AgNW-MXene, and AgNW-MP films with different mass ratios [82]. (**d**) Optical contrast ΔE* in relationship with the thickness of the electrochromic layer and the wt% V_2_O_5_ in the PEDOT:PSS ink [62]. (**e**) Refractive index versus wavelength of different PEDOT:PSS samples [114].

### 2.5. Spectral Scattering Property

As previously mentioned, a flat and dense PEDOT film minimizes light scattering. The application of optical scattering in this context is primarily seen in the use of Raman spectroscopy for characterizing PEDOTs. Raman spectroscopy is a widely used technique to study the molecular structures and vibrational characteristics of PEDOTs, offering insights into their conformation and crystallinity. This helps elucidate the relationship between the molecular structure, conjugation length, doping conditions, and properties of PEDOTs. The Raman spectrum of PEDOT features characteristic peaks in the range of 1400–1500 cm^−1^ [115], the positions and intensities of which depend on the molecular structure and bonding states of PEDOT. Such data provide critical information for material design and synthesis. Moreover, when combined with electrochemical techniques, Raman spectroscopy can enable in situ characterization of molecular structural changes during electrochemical reactions, providing insights into reaction mechanisms. Currently, wavefront shaping techniques, optical phase conjugation, and transmission matrix methods have demonstrated excellent results in optimizing the propagation path of light and achieving precise control over light. These methods are gradually being applied to the field of conductive materials. Although the optical scattering properties of PEDOT films have not yet been extensively studied, the use of advanced photomodulation techniques presents a promising avenue for future research. These techniques are expected to enable accurate exploration of the optical scattering performance of PEDOT films under various conditions and provide a detailed analysis of the scattering behavior when light interacts with PEDOT films. This will contribute to a deeper understanding of their optical scattering properties. Furthermore, the rapid advancements in artificial intelligence (AI) technology open new opportunities for studying the optical scattering properties of PEDOT films. By integrating AI with wavefront shaping techniques, machine learning algorithms can be employed to predict and optimize scattered light fields, enabling more comprehensive and in-depth investigations of the optical scattering characteristics of PEDOT films. This integration lays a robust theoretical and practical foundation for innovative applications of PEDOT films in diverse fields, including optoelectronic devices.

### 2.6. Modification Methods for Optical Properties

To optimize the optical properties of PEDOTs, researchers have adopted a variety of strategies. Effective doping/dedoping allows reversible regulation of PEDOT’s oxidation and reduction states, enabling electrical transport tuning and significantly influencing optical properties. PEDOT:PSS can also be modified using additives such as polar solvents, inorganic salts, acids, bases, ionic liquids, and redox agents, which alter conformational arrangements and oxidation levels, thereby affecting optical properties [116]. Additional control over optical properties can be achieved by tuning molecular structures, dopant types, and concentrations.

Compositing PEDOTs with other materials—such as metal nanoparticles [117,118,119], transition metal oxides [120,121], and carbon quantum dots [122]—further enhances their optical properties and broadens their application potential. PEDOT films can be prepared using techniques such as spin-coating [123], inkjet printing [124], 3D printing [125], photolithography [126], screen printing [127], and electrochemical deposition [128], all of which impact their optical properties. For instance, optimizing spin-coating parameters can yield PEDOT films with high light transmittance and low surface roughness, while electrochemical deposition produces films with uniform thickness and excellent conductivity. The choice of printing methods and processes plays a crucial role in determining the quality of PEDOT films and the performance of corresponding devices. For instance, 3D printing, also known as additive manufacturing, has emerged as a transformative approach for fabricating complex and customized optoelectronic devices. Utilizing PEDOT-based inks in 3D printing offers distinct advantages, including precise control over deposition location and shape, enabling the creation of intricate 3D structures for optoelectronic applications [129,130]. Specifically, 3D printing can enhance light harvesting and scattering effects by constructing complex geometric structures, thereby improving overall light extraction efficiency [14].

However, challenges remain in optimizing interlayer bonding strength and printing speed to ensure the electrical conductivity and mechanical stability of PEDOT films [131]. Additionally, ink viscosity and drying rate control are critical issues: excessively high viscosity can clog printing nozzles, while excessively low viscosity can lead to pattern spreading and reduced resolution. In contrast, screen printing, while offering lower resolution, is cost-effective and well suited for large-area applications where high resolution is not a primary requirement [132]. Screen printing produces relatively thick PEDOT films that may improve electrical conductivity but could also increase light absorption and scattering, potentially affecting the device’s optical performance. Understanding and optimizing the interplay between printing techniques, film properties, and device performance is essential for advancing high-performance PEDOT-based optoelectronic devices.

The regulation of PEDOTs’ optical properties is not confined to a single parameter or limited to the interplay of electrical and mechanical properties. For example, Xie et al. [133] synthesized an intrinsically self-healing conductive film by integrating Hydroxymethyl EDOT (EDTM) units into the PEDOT backbone and combining it with thermally self-healing polyurethane (PU-DA) containing reversible Diels–Alder groups. The P(EDOT-co-EDTM)/PU-DA copolymer film, prepared by spin-coating onto ITO glass, exhibited outstanding electrochromic performance, including a visible light contrast of 47.24%, a response time of 0.8 s, and a coloring efficiency of 324.9 cm^2^ C^−1^. After 100 cycles, its visible light contrast remained above 93%. Similarly, Ling et al. [134] used a one-pot method to electrochemically deposit a hybrid film of PEDOT:PSS and WO_3_ onto ITO glass. By controlling deposition conditions to adjust morphology, they achieved an optical contrast of 45%, surpassing that of individual PEDOT:PSS or WO_3_ films of the same thickness.

## 3. Electrical and Electrochemical Properties of PEDOTs and Structural Tunability

The exceptional electrical properties of PEDOTs primarily stem from the π-π conjugation system within their molecular chains, which facilitates the free movement of electrons, aided by anion doping. This unique structure imparts PEDOTs with excellent electrical conductivity, typically ranging from hundreds to thousands of S cm^−1^, depending on factors such as molecular structure, doping levels, preparation methods, and processing conditions [135]. These attributes enable PEDOTs to find broad applications in organic electronic devices, including ECDs, OSCs, OLEDs, transparent electrodes, antistatic coatings, sensors, and more [136,137,138,139].

In addition to their electrical conductivity, PEDOTs are being increasingly studied for their thermoelectric and piezoelectric properties, revealing their potential in energy conversion and intelligent sensing applications [140,141,142]. Furthermore, PEDOTs exhibit noteworthy electrochemical redox activity, making them highly promising for use in batteries, supercapacitors, thermoelectric devices, and ECDs [143,144,145]. This electrochemical activity, combined with their remarkable stability over wide potential ranges and high charge mobility within polymer chains, underscores their utility in energy storage and conversion technologies. It is important to note that PEDOTs’ electrical properties are intricately interconnected with other attributes, such as their optical characteristics, mechanical behavior, self-healing capabilities, biocompatibility, and surface/interface features. These properties often exhibit competitive or synergistic relationships, providing significant opportunities for optimization. Consequently, enhancing and tuning PEDOTs’ electrical properties remains a central focus in this field.

Given the availability of several comprehensive reviews on the electrical and thermoelectric optimization of PEDOT:PSS films [135,146,147], this discussion provides a brief overview of the fundamental electrical mechanisms of PEDOTs and their primary regulation methods. It adopts a broader perspective, encompassing but not limited to PEDOT:PSS.

### 3.1. Electrical Properties

#### 3.1.1. Modification Methods to Enhance Electrical Conductivity

In the molecular structure of EDOT, the 3- and 4-positions on its thiophene ring are substituted with side groups, restricting polymerization to the 2- and 5-positions. This results in a linear, non-crosslinked PEDOT backbone with minimal conjugated defects. The ether substituents on the thiophene ring not only lower the oxidation potential of EDOT—facilitating polymerization—but also reduce the oxidation potential of PEDOT, enhancing its stability during redox (doping and dedoping) cycles [148]. This molecular structure establishes a conjugated π-electron system in PEDOT, enabling efficient carrier transport along the molecular chains and imparting electrical conductivity. The linear, non-crosslinked structure with fewer defects further facilitates carrier mobility.

However, the strong π-π electronic interactions between PEDOT backbones result in insolubility and non-meltability, creating processing challenges and limiting certain applications [135]. This challenge was addressed by doping PEDOT with the water-soluble polymer electrolyte PSS, leading to the development of PEDOT:PSS. PEDOT:PSS has gained significant attention due to its simple preparation, scalability, and excellent optical and electrical properties. Commercially, PEDOT:PSS is available as an aqueous dispersion with electrical conductivity ranging from 10^−1^ to 100 S cm^−1^, depending on the PEDOT-to-PSS ratio. In water, PEDOT:PSS forms a core–shell structure, where hydrophilic PSS acts as a shell surrounding a PEDOT-rich conductive core, a structure that persists even in dried films. However, the insulating PSS shell hinders charge transport, resulting in relatively low electrical conductivity. Additionally, as a soft polymer, PSS tends to adopt a coiled conformation, which restricts the PEDOT chains or segments connected through Coulombic attraction, further limiting charge mobility. The current mechanism for enhancing conductivity, as illustrated in Figure 3, relies on inducing phase separation of PEDOT:PSS chains within PSS, reducing Coulombic interactions between positively charged PEDOT and negatively charged PSS dopants through shielding, and transforming PEDOT chains from a helical to a linear conformation [149,150].

Alternative strategies to improve performance include increasing the proportion of PEDOT, enhancing crystallinity and oxidation levels, reducing the presence of insulating PSS, and promoting more distinct phase separation between PEDOT and PSS. These improvements can be achieved through various chemical and physical methods, such as additive mixing, solvent post-treatment, light exposure, and heat treatment (Figure 3).

**Figure 3 molecules-30-00179-f003:**
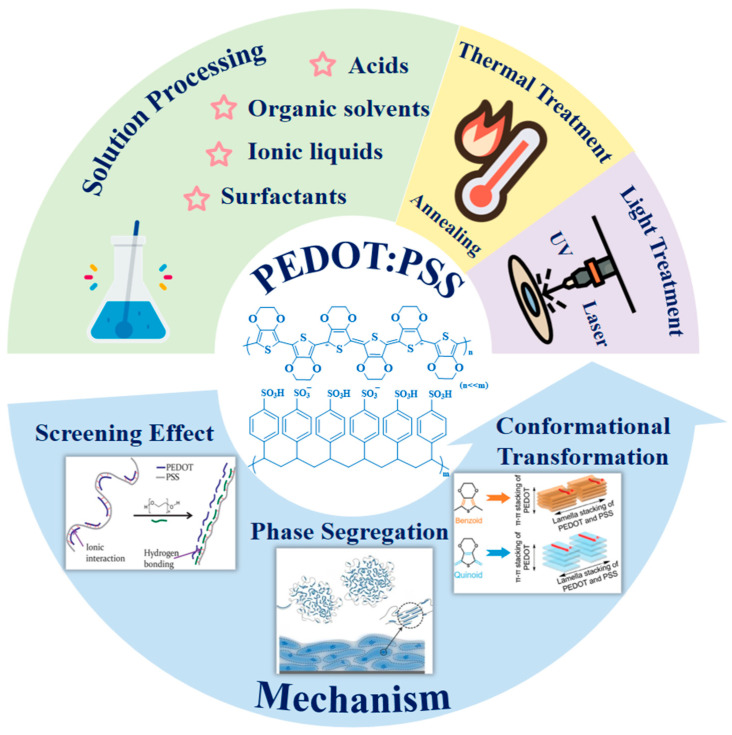
Schematic diagram illustrating the key methods for regulating electrical conductivity in PEDOT:PSS and their underlying mechanisms [149,150,151].

Additive treatment is a technique used to modify the electronic structure and molecular chain conformation of PEDOT:PSS by introducing external substances prior to film processing. Commonly utilized additives include organic solvents [150,152], ionic liquids [153], surfactants [154], and others. Among these, the addition of polar organic solvents is not only the earliest proposed but also the most widely adopted method for significantly enhancing electrical conductivity. Examples of such solvents include dimethyl sulfoxide (DMSO), ethylene glycol (EG), *N*,*N*-dimethylacetamide (DMAC), N,N-dimethylformamide (DMF), methanol (MeOH), D-sorbitol, xylitol, glycerol, and more [155,156,157,158,159,160,161,162,163,164,165,166,167,168,169,170,171,172,173]. Polar solvents with high dielectric constants exert strong screening effects on counterions and carriers, thereby reducing the Coulombic interaction between positively charged PEDOT and negatively charged PSS dopants. These solvents also induce conformational changes in the PEDOT:PSS chains, promoting better alignment and increasing carrier mobility within the molecular structure and bulk phase film [135,174]. Consequently, the screening effect of polar solvents plays a pivotal role in enhancing the electrical conductivity of PEDOT:PSS. Among these solvents, DMSO has been shown to induce the most pronounced effect, improving conductivity by up to two orders of magnitude [116].

Ionic liquids (ILs) possess unique properties, including low vapor pressure, high thermal and chemical stability, and excellent solubility. As modifiers for PEDOT:PSS, ILs primarily promote phase separation between PEDOT and PSS by inducing ion exchange processes, reducing Coulombic interactions, and improving the crystallinity and π-π stacking order of PEDOT. These changes optimize charge transport channels, leading to significant enhancements in electrical conductivity [175]. Based on these properties, several ILs have been widely studied as additives for PEDOT:PSS, including 1-ethyl-3-methylimidazoles (EMIM) [176,177,178], bis(trifluoromethane)sulfonamides (TFSI) [179], and 1-ethyl-3-methylimidazolium chloride (EMIC) [180]. These additives induce conformational changes in PEDOT molecules, transitioning their structure from benzene-type to quinone-type configurations. This planarization of PEDOT chains increases interchain stacking density, facilitating faster charge transfer and significantly improving electrical conductivity [135].

Additionally, the cations and anions in ILs can act as charge transport media, reducing interfacial barriers across different molecular chains and further accelerating charge transport within PEDOT:PSS films [181,182,183]. IL treatment also improves the morphology of PEDOT:PSS films, making them more uniform and dense, which reduces defects and pores and contributes to enhanced conductivity [184]. These ionic characteristics also improve the performance of composite films in photoelectronic devices. For example, the fluoride ionic liquid Zinc di[bis(trifluoromethanesulfonyl)imide] (Zn(TFSI)_2_) has been shown to induce phase separation between PEDOT and PSS, forming a network-like structure. This results in an electrical conductivity as high as 4115 S cm^−1^, comparable to ITO on polyethylene glycol terephthalate (PET) substrates [185]. Furthermore, these films demonstrate high optical transmittance (over 80% in the 400–900 nm range) and flexibility, making them ideal for flexible perovskite solar cells, where they contribute to significant improvements in photovoltaic conversion efficiency. PEDOT:PSS is widely employed as an HTL in OSCs due to its favorable carrier mobility, tunable work function, and photon utilization properties. These features enable efficient collection and transport of photogenerated holes, thereby minimizing recombination losses and enhancing the PCE of OSCs [186,187]. PEDOT:PSS films also form excellent interfacial contact with light-absorbing layers in various solar cells, including those based on perovskites, phthalocyanines, perylene tetracarboxylic diimides, and thiophene-based polymers. This outstanding interfacial compatibility reduces interfacial resistance, improves carrier collection efficiency, and minimizes charge recombination at the interface, thereby boosting the overall performance of OSCs [188].

At the molecular level, holes primarily migrate along the π-π stacked PEDOT backbone, while PSS chains act as insulating dispersants. Optimizing the preparation and modification of PEDOT:PSS can further enhance its application effectiveness in OSCs [189]. For instance, crosslinking with ethylenediamine-functionalized graphene (EDA-FG) can rearrange PEDOT:PSS molecules, improving surface roughness, crystallinity, and wettability of the PEDOT:PSS layer. This, in turn, facilitates the growth of high-quality perovskite crystals [190]. Consequently, OSCs utilizing this bilayer HTL have demonstrated exceptional photoelectric performance, achieving a maximum PCE of 17.66% and stable photovoltaic behavior for over 500 h.

Surfactants demonstrate significant potential in regulating the electrical conductivity of PEDOT:PSS beyond their common role in solubilization. Their amphiphilic nature, comprising hydrophilic and hydrophobic groups, enables them to arrange themselves at solution surfaces or interfaces, thereby reducing surface and interfacial tension. Commonly used surfactants for PEDOT:PSS include non-ionic surfactants such as polyethylene glycol (PEG) [145,191], polyvinyl alcohol (PVA) [192], Triton X-100 [193,194,195], Zonyl [196,197,198], and the Tween series (e.g., Tween 20, Tween 80) [199,200], as well as anionic surfactants like sodium dodecyl sulfate (SDS) [201,202,203] and glycerol monostearate (GMS) [204]. These surfactants reduce surface tension between PEDOT:PSS particles, enabling the formation of a more uniform dispersion in water [116]. This minimizes particle aggregation and promotes the extension of PEDOT chain segments, enhancing the number of conductive pathways and improving electrical conductivity.

Additionally, surfactants can adsorb onto the surface of PEDOT:PSS particles, modifying their surface composition and charge distribution. This improves the interface contact between PEDOT:PSS and electrodes or other functional layers, reducing contact resistance and further enhancing electrical conductivity [205]. Moreover, surfactants can regulate the work function of PEDOT:PSS to better align with other electronic materials, thereby improving overall device performance [206].

Post-solvent treatment methods, such as solution dripping, soaking, and polar solvent vapor annealing, are commonly employed to enhance the properties of PEDOT:PSS films [116]. These treatments can modify the conformational arrangement of PEDOT:PSS chains and remove the insulating PSS phase, thereby improving the π-π stacking between PEDOT chains and optimizing both the crystallinity and crystal morphology of PEDOT [135]. Inorganic acids are frequently used as solvents for these treatments, including hydrochloric acid (HCl) [207], sulfuric acid (H_2_SO_4_) [208], nitric acid (HNO_3_) [209], phosphoric acid (H_3_PO_4_) [210], chloroplatinic acid, acetic acid, propionic acid, methanesulfonic acid, and others. Among these, concentrated H_2_SO_4_ has proven to be one of the most effective methods for significantly increasing electrical conductivity, reaching values as high as 4380 S cm^−1^ [208]. For example, Liu et al. [211] utilized vacuum-assisted filtration to prepare PEDOT:PSS/aramid nanofiber (ANF) composite films. After post-treatment with H_2_SO_4_, the electrical conductivity of the composite films increased by 20–25 times compared to untreated films. However, the use of strong acids, while highly effective, poses challenges such as substrate corrosion, which can compromise the overall performance and long-term stability of electronic devices.

Heating and photothermal post-treatments are effective strategies for adjusting the electrical conductivity of PEDOT:PSS films, as conductivity is strongly influenced by film morphology and its chemical and physical structure. These treatments are typically implemented through film annealing processes. Thermal treatment rearranges PEDOT segments, reducing molecular chain entanglement and improving chain orientation, thereby facilitating charge transport within the film [135]. Additionally, thermal treatment promotes phase separation between PEDOT and PSS, resulting in the formation of more continuous conductive channels in PEDOT-rich regions, further enhancing electrical conductivity.

Thermal treatment can also be combined with other methods to achieve even greater improvements. For instance, Chen et al. [151] incorporated ice crystallization confinement effects to significantly enhance the electrical conductivity of PEDOT:PSS films, achieving values up to 2564 ± 142 S cm^−1^. In this process, water crystallization facilitates phase separation between PEDOT and PSS. During the ice crystal formation stage, hydrophilic PSS adsorbs onto the surface of ice crystals through hydrogen bonding, while PEDOT is confined within the channels formed between the ice crystals, creating a preliminary conductive network. Subsequent annealing further promotes π-π stacking among PEDOT molecular chains, enhancing crystallinity and conductivity.

However, excessively high temperatures can lead to decomposition or over-drying of PEDOT:PSS films, necessitating precise control over temperature and duration during thermal treatment. For example, Jeong et al. [212] subjected PEDOT:PSS films to hydrothermal treatment under specific conditions (temperature > 61 °C and relative humidity > 80%). This treatment enhanced the electrical conductivity of the films to 125.367 S cm^−1^. This simple and biocompatible method induced structural rearrangement within the PEDOT:PSS films, enhancing chain bridging, promoting PEDOT crystallization, and reducing the PSS content. Under optimized conditions, PEDOT chains transitioned from a coiled state to a linear or extended state, improving overall crystallinity. At relative humidity levels exceeding 80%, a water meniscus layer formed, creating a shielding effect between PEDOT and PSS, further reducing Coulombic attraction and enhancing conductivity. Electrodes prepared through the hydrothermal treatment of PEDOT:PSS can effectively monitor joint movements and skin temperature, as well as measure ECG signals. Unlike organic solvent doping and post-treatment methods, this approach leaves no undesirable residue. These biocompatible, conductivity-enhancing methods demonstrate significant potential for a wide range of biomedical applications.

Light treatment can also be applied to modulate the electrical conductivity of PEDOT:PSS films, which involves irradiating the film with ultraviolet (UV), visible, or laser light to enhance the electrical properties. This process can initiate photochemical reactions within the PEDOT chains, such as crosslinking, oxidation, or reduction, leading to alterations in the material’s chemical structure. The energy provided by light promotes molecular migration and serves as a critical driver for phase separation between PEDOT and PSS [213,214]. This phase separation facilitates the agglomeration of PEDOT-rich particles and enhances the conjugation of PEDOT chains, which collectively improve the electrical conductivity of the material [215,216,217].

The effectiveness of light treatment depends on various factors, including the type of light source, light intensity, exposure duration, and film thickness. For instance, Ding et al. [218] demonstrated precise control over the microstructures and electrical properties of PEDOT:PSS films using pulsed CO_2_ laser irradiation. By optimizing irradiation parameters such as laser fluence and scanning speed, they achieved a peak electrical conductivity of 59.4 S cm^−1^ at a laser fluence of 27.85 mJ cm^−2^. This significant enhancement was primarily attributed to internal microstructural changes induced by laser treatment, including the re-bonding of PEDOT and PSS molecules and the enrichment of PEDOT on the film surface. These modifications reduced the hopping distance of charge carriers and increased both the quantity and quality of conductive channels within the film.

In short, modulating the electrical conductivity of PEDOT:PSS is a complex and nuanced process that requires careful consideration of multiple factors. By employing rational strategies, such as composite system construction before film formation or post-treatment of dried films, the electrical conductivity of PEDOT:PSS-based films can be significantly enhanced. These advancements lay a robust foundation for the widespread application of PEDOT:PSS in electronic devices. On the other hand, structural derivation, which is the most fundamental approach to regulation, offers potential improvements in the processability of PEDOT. However, modifications such as side-group substitution or main-chain derivation typically lead to a decrease in electrical conductivity. This reduction occurs because these modifications can partially disrupt the conjugated chain and weaken carrier transport capabilities. As a result, most practical applications in this field continue to rely on unmodified PEDOT or PEDOT:PSS. Nonetheless, as highlighted in the optical section, structural derivation undoubtedly introduces a wide range of additional functionalities [219,220]. These include, for example, color-tuning capabilities for electrochromism, recognition and adsorption functionalities for biosensors, system crosslinking, and interface enhancement for applications in OSCs and hydrogel electronics. Given the vast number of molecular structures and synthetic derivatives available—numbering in the hundreds—a comprehensive analysis of PEDOT molecular derivatives will be specifically addressed in a subsequent review.

While strategies for regulating the electrical conductivity of PEDOTs demonstrate great potential for enhancing material properties, they are accompanied by several challenges and hazards across production, processing, and application stages.

First, in terms of solvent doping, the use of a large quantity of organic solvents poses a significant concern. These solvents vary in volatility, stability, and their interaction mechanisms with PEDOT:PSS, leading to unpredictable miscibility and system stability. Mixing different solvents may trigger chemical reactions or physical changes, adversely affecting PEDOT:PSS performance. This increases uncertainty in the production process and impacts the final product’s quality. Furthermore, most doping formulations need to be used immediately and cannot be stored for extended periods, limiting their scalability. For instance, adding PEG significantly increases solution viscosity, creating processing challenges such as difficulties in uniform coating and precise patterning through conventional printing or coating methods. DMSO, a commonly used solvent, enhances conductivity effectively but has strong toxic side effects. Its residues can pose potential health risks, particularly in applications involving close human contact, such as biomedical sensors and wearable electronics. Additionally, DMSO’s high boiling point raises the film-forming temperature, increasing energy consumption and restricting its use in biological systems where high temperatures can damage biologically active substances or tissues.

Reprocessing methods involving acids and bases also present significant safety and environmental concerns. Strongly corrosive acids and alkalis can damage equipment, harm operators, and degrade the production environment. In industrial settings, long-term use of such treatments often leads to equipment corrosion, higher maintenance costs, and increased replacement frequency. Moreover, the leakage or volatilization of these substances into the environment results in ecological pollution, conflicting with sustainable development goals.

To address the hazards of solvent doping, the development of low-toxicity, small-molecule, low-boiling-point polar solvents to replace DMSO is crucial. Alternative solvents must exhibit good stability and appropriate interactions with PEDOT:PSS. Similarly, the development of milder reprocessing methods is necessary to mitigate the limitations of highly corrosive acid and alkali treatments, such as concentrated sulfuric acid.

Mechanical stresses during use often generate cracks and defects in PEDOT-based materials, compromising their electrical properties and limiting their application in flexible and portable devices requiring durability. Self-healing techniques are being explored as a potential solution [221,222], though the self-healing process often competes with electrical performance, necessitating a balance between these properties for practical applications.

#### 3.1.2. Modification Methods to Enhance Thermoelectricity

Unlike photoelectronic devices such as OSCs, TE materials enable the direct conversion between thermal and electrical energy through the transport of intrinsic carriers. These materials hold significant application potential in energy utilization, flexible wearable generators, temperature sensors, and other related fields [223,224,225]. The performance of TE materials is primarily characterized by the Seebeck effect and the Peltier effect. The Seebeck effect refers to the generation of an electric potential difference within a material when a temperature gradient exists across its ends, thereby converting thermal energy into electrical energy. Conversely, the Peltier effect is the reverse process, where an electrical current passing through the material creates a temperature difference across its ends, converting electrical energy into thermal energy [226,227].

TE performance is evaluated using a dimensionless figure of merit (*ZT*), expressed by the equation:
*ZT* = (*σ**S*^2^*T*)/*κ*(1)
where *σ* is the electrical conductivity, *S* is the Seebeck coefficient, *T* is the absolute temperature, and *κ* is the thermal conductivity. The term *σS*^2^ is known as the power factor (PF). For practical applications, the *ZT* value needs to be as high as possible, with a target of at least 1.0.

PEDOT has emerged as a promising material for TE applications due to its low thermal conductivity and high electrical conductivity after doping, along with other exceptional properties [228,229,230]. In PEDOT, the primary charge carriers are electrons and holes, and their transport properties are critical to its TE performance [231]. Similarly to electrical conductivity modulation, TE properties can be optimized through techniques such as doping and compositing. These methods adjust carrier concentrations and transport properties, which are reflected in the overall improvement of TE performance.

Doping with various counter anions is one of the effective strategies for altering the TE properties of PEDOT [229]. However, neither PEDOT itself, its derivatives, nor other CPs exhibit substantial advantages in thermoelectric performance. In contrast, PEDOT:PSS films are widely recognized for their significant potential in advancing organic TE materials, attributed to their high electrical conductivity, superior thermal stability, and other advantageous characteristics [231,232,233].

Nevertheless, the inherent trade-off between electrical conductivity and the Seebeck coefficient complicates the optimization of TE properties. This relationship requires a more intricate approach compared to solely enhancing electrical conductivity. For instance, Tu et al. [234] developed a physicochemical dedoping method that effectively regulates the TE performance of PEDOT:PSS films. This approach involves promoting the aggregation and phase separation of PEDOT molecules through the addition of dimethyl sulfoxide (DMSO), followed by dedoping treatment using a DMSO/sodium thiosulfate (Na_2_SO_3_) salt solution.

DMSO plays a dual role: its polar groups interact with PSS chains via hydrogen bonding and engage in dipole–dipole or dipole–charge coupling with PEDOT chains. These interactions extend the PSS molecular chains while rearranging the PEDOT chains, creating a more ordered structure. The subsequent dedoping process with DMSO/salt solution selectively removes excess non-conductive PSS chains, reducing spatial hindrance and enabling the PEDOT and PSS chains to adopt a more linear and extended alignment. This combined treatment significantly improves both the electrical conductivity and the Seebeck coefficient of the material. Notably, the power factor achieved a substantial enhancement, reaching 105.2 µW m^−1^ K^−2^ when the volume ratio of DMSO to Na_2_SO_3_ solution was 1:10 (Figure 4a).

Organic–inorganic hybrid composites have emerged as a promising direction in the thermoelectric (TE) field [235,236]. The formation of organic–inorganic heterogeneous interfaces can hinder the transport of low-energy carriers while elevating the average energy of carriers within the system. This mechanism enhances the Seebeck coefficient of composite films [233,237]. Inorganic TE materials, such as telluride alloys (e.g., BixSbyTez [238] and Bi_2_Te_3_ [239]) and certain two-dimensional (2D) materials like SnSe [240] and MoS_2_ [241], have proven instrumental in improving the TE performance of PEDOT:PSS composites. Compared to mixed-structure devices, double-layer structures of PEDOT/Te films significantly enhance both the electrical conductivity and the Seebeck coefficient of PEDOT:PSS. The semi-metallic nature of PEDOT:PSS and Te enables the formation of an ohmic contact interface, which is conducive to charge transport [242]. Additionally, capacitance–voltage (C-V) measurements confirm the accumulation of interface charges in such devices (Figure 4b). As the temperature increases, the Fermi level approaches the conduction band, leading to a higher number of high-energy carriers at the interface. This dynamic effectively enhances the Seebeck coefficient of PEDOT:PSS, contributing to its overall TE performance.

Optical regulation has also emerged as an innovative approach for optimizing the thermoelectric (TE) performance of PEDOT:PSS. Under illumination, electrons in PEDOT:PSS absorb light energy and transition from the valence band to the conduction band, generating free carriers [231]. These photogenerated carriers impact the Seebeck coefficient by altering the carrier concentration, which shifts the Fermi level and affects the energy filtering effect during TE conversion [229,243]. For example, Wang et al. [244] demonstrated the effectiveness of femtosecond laser processing in regulating the electrical conductivity, Seebeck coefficient, and power factor of PEDOT:PSS films. By adjusting laser irradiation parameters—without any chemical doping or additional treatment—they achieved significant performance improvements. Femtosecond laser-induced carrier excitation and the delocalization of the conjugated backbone in PEDOT chains enhanced electron transport properties within the film (Figure 4c). At an optimal laser irradiation intensity of 3.42 mJ cm^−2^, the electrical conductivity of PEDOT:PSS films increased from 1.2 S cm^−1^ to 803.1 S cm^−1^, with the power factor reaching a peak value of 19.0 µW m^−1^ K^−2^, showcasing the immense potential of this technology (Figure 4d). The high polaron concentration and carrier mobility contributed significantly to these enhancements, improving both electrical conductivity and the material’s figure of merit (ZT).

In another example, Zhang et al. [245] introduced the concept of polaron interface entropy engineering, utilizing photochromic diarylethene (DAE) molecules and UV light modulation to optimize TE performance. By doping PEDOT:PSS with DAE molecules, UV light irradiation induced a photochromic reaction that transitioned DAE molecules from an open-ring structure to a closed-ring structure. The closed-ring molecules formed planar coupling interface states with PEDOT chains via similar C–C=C–C bonds, enabling resonant coupling between them. This coupling created new polaron interface states and increased polaron interface entropy, offering additional carrier occupation sites and enhancing TE performance. Under UV light, the thermopower of PEDOT:PSS films increased significantly with higher DAE concentrations, peaking at approximately 135.5 μV K^−1^ at a DAE concentration of 38 wt.%. Meanwhile, the electrical conductivity decreased only slightly (by ~10%) under UV modulation, while the power factor improved by 93 times, reaching 527 µW m^−1^ K^−2^ (Figure 4e). These findings highlight the transformative potential of optical regulation for TE applications in PEDOT:PSS.

To conclude, PEDOTs, particularly PEDOT:PSS, have established themselves as some of the most prominent organic thermoelectric materials, demonstrating remarkable performance. Despite ongoing challenges in achieving an optimal balance among the various physicochemical properties that contribute to a high ZT value, the potential of these materials remains immense. In the field of wearable biosensors, it can enable body temperature monitoring, energy harvesting, bioelectric signal detection, and energy supply. Applications include monitoring body temperature in sports, fitness, and medical rehabilitation scenarios, detecting bioelectric signals in long-term cardiac monitoring devices, and accurately identifying complex hand movements. Additionally, in the field of implantable medical devices, it can be utilized for deep tissue temperature monitoring, energy self-sufficiency, nerve stimulation, and energy supply. Their promising applications in microelectronics, lightweight and flexible devices, wearable technologies, and multifunctional electronic systems are highly anticipated.

**Figure 4 molecules-30-00179-f004:**
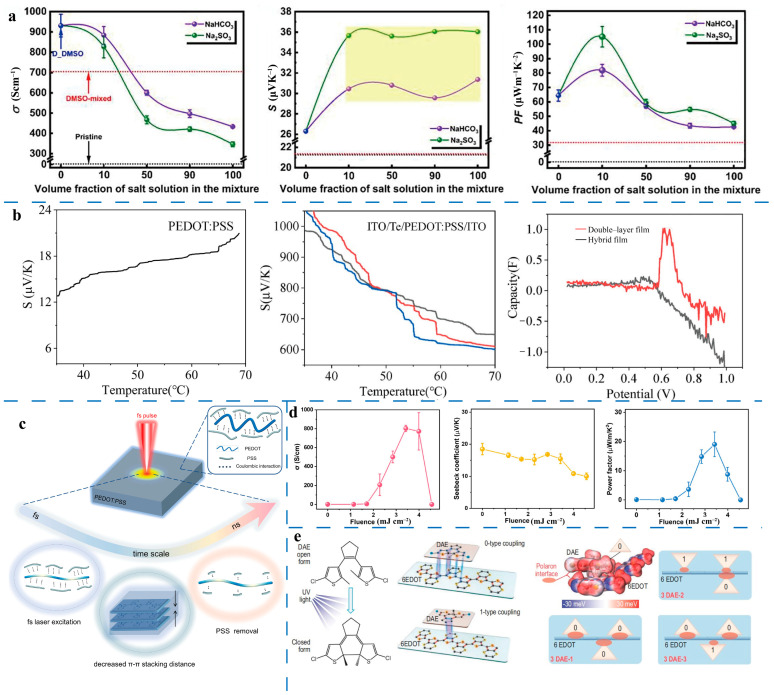
(**a**) TE properties of PEDOT:PSS films under different treatment conditions in terms of electrical conductivity, Seebeck coefficient, and power factor [234]. (**b**) Seebeck coefficient PEDOT:PSS and PEDOT:PSS/Te double-layer film changing with temperature, and C-V characteristic of PEDOT:PSS/Te hybrid and double-layer films under room temperature (Three tests were tested on devices of the same construction) [242]. (**c**) Schematic illustration of the mechanism for enhanced electrical conductivity in PEDOT:PSS through femtosecond pulsed laser treatment [244]. (**d**) Electrical conductivity, Seebeck coefficient, and power factor of PEDOT:PSS films after annealing with different laser fluence [244]. (**e**) Polaron interfacial occupied entropy engineering of photochromic DAE molecules [245].

#### 3.1.3. Modification Methods to Enhance Piezoelectricity

The piezoelectric effect refers to a material’s ability to convert mechanical stress into electrical energy [246,247]. This phenomenon arises primarily from the non-centrosymmetric crystal structure of piezoelectric materials. When subjected to external forces, these materials experience changes in internal charge distribution, resulting in the redistribution or movement of charges and subsequently generating a potential difference or current [248,249,250]. Piezoelectric materials have a wide range of applications, including sensors, transducers, actuators, and other devices. They are particularly crucial in fields such as energy harvesting [251,252,253], neural tissue engineering [254,255], robotics [256], catalysis [257,258], medical healthcare [259,260], and electronic information systems [261].

Common types of piezoelectric materials include: single crystals such as quartz, lithium niobate, lithium tantalate, etc. [262]; ceramics such as zirconate titanate series, barium titanate (BaTiO_3_), etc. [263,264,265]; metal oxides [266]; organic piezoelectric materials such as polyvinylidene fluoride (PVDF), polyacrylonitrile (PAN), etc. [267,268]; and polymers [269]. While PEDOT and PEDOT:PSS do not exhibit intrinsic piezoelectricity, their exceptional electrical conductivity allows for efficient charge transport. PEDOTs possess a conjugated structure that allows them to exhibit excellent electrical conductivity in the oxidized state—an essential feature that differentiates them from conventional piezoelectric ceramic materials. This property significantly enhances the overall material’s conductivity, facilitating efficient charge transfer and distribution generated by the piezoelectric effect. This improves the electrical connectivity of electronic devices and enhances the performance of piezoelectric materials. Additionally, incorporating PEDOT:PSS can alter the crystalline structure of certain piezoelectric polymers (e.g., PVDF), promoting the piezoelectric effect and increasing the film’s sensitivity to force, thereby further enhancing its piezoelectric properties. As a result, they are frequently composited with piezoelectric materials for use in applications such as piezoelectric sensors, energy harvesters, and related fields [270,271,272,273,274].

Poly(vinylidene fluoride) (PVDF) and its copolymers are widely utilized in triboelectric and piezoelectric nanogenerators due to their excellent chemical resistance, elasticity, biocompatibility, mechanical properties, and controllable ferroelectric characteristics. As semi-crystalline polymeric materials with three crystalline phases (α, β, and γ), the piezoelectric performance of PVDF and its derivatives is primarily determined by the β-phase content [268]. Leveraging these properties, Li et al. [275] utilized interfacial engineering to fabricate a P(VDF-TrFE)/PEDOT:PSS composite film by integrating DMSO-doped PEDOT:PSS with P(VDF-TrFE) at the interface to form a network interconnection interface (NII) (Figure 5a). This composite exhibited a piezoelectric coefficient of 86 pC N^−1^ and a pyroelectric coefficient as high as 95 μC m^−2^ K^−1^. The P(VDF-TrFE)/PEDOT:PSS composite film demonstrated ultra-high sensitivity in pressure and temperature sensing, achieving values of 2.2 V kPa^−1^ (in the pressure range of 0.025–100 kPa) and 6.4 V K^−1^ (in the temperature variation range of 0.05–10 K), respectively (Figure 5b). Mechanical reliability testing of PEDOT:PSS as a flexible electrode showed an increase in resistivity from 68 Ω cm^−1^ to 85.5 Ω cm^−1^ after 10,000 bending cycles, highlighting the electrode’s excellent durability. AFM observations revealed that the PEDOT:PSS electrode surfaces remained smooth and free of notch defects before and after bending, indicating that the NII retained its integrity during mechanical wear. This ensured the stability of the piezoelectric and thermoelectric properties of the composite films during prolonged use.

In long-term durability tests involving 1000 loading cycles, the piezoelectric response curve of the composite film remained stable, with an output voltage of approximately 220 V. Similarly, the thermoelectric output voltage remained constant at around 62 V after more than 1000 heating cycles at 10 K. These findings further demonstrate the stability of the NII under mechanical wear conditions. The superior mechanical coupling effect resulting from the network interconnection between PEDOT:PSS and P(VDF-TrFE) facilitated a positive response to pressure-induced strain. Additionally, the network interconnections increased the contact area, reduced the composite film’s total resistance, and enhanced its piezoelectric properties.

Wu et al. [276] developed a flexible PEDOT:PSS-PVDF(HFP) film sensor through a solvent-induced strategy, combining PEDOT:PSS and poly(vinylidene fluoride-hexafluoropropylene) (PVDF(HFP)) for detecting triple signals: strain, pressure, and humidity (Figure 5c). The DMSO-doped PEDOT:PSS-PVDF(HFP) film demonstrated an electrical conductivity of 800 S cm^−1^, a piezoelectric constant of 9.30 pC N^−1^, and humidity response/recovery times of 1.15 s/0.42 s. These outstanding properties were attributed to the polar solvent DMSO, which removed insulating PSS, transformed the PEDOT conformation, and increased the β-phase content in PVDF(HFP). The high molecular weight polymer chain mixing, deformation capacity of the conductive PEDOT:PSS path, piezoelectric effect of PVDF(HFP), and differential asymmetric expansion due to the bilayer composite structure’s varied wettability all contributed to the film’s multi-signal response capability. In the context of strain sensing, PEDOT:PSS-PVDF(HFP) film can be directly attached to various parts of the human body, such as fingers, wrists, elbows, throat, and face. It can monitor changes in electrical signals during different movements, including finger bending, wrist rotation, elbow flexion, elbow extension, swallowing, and facial expressions. These sensors reliably and stably acquire human body movement data without causing adverse effects. Long-term tests, such as the 24-h vertical tensile test, show that the PEDOT:PSS-PVDF(HFP) film retains its initial state and conductivity. This demonstrates that its biocompatibility plays a crucial role in maintaining sensor stability during prolonged use, ensuring its reliability and safety for applications such as long-term health monitoring.

Beyond all-organic composite films, PEDOT and PEDOT:PSS can also be incorporated into inorganic piezoelectric systems, leveraging their high electrical conductivity (p-type) and mechanical flexibility. For instance, zinc oxide (ZnO), with its hexagonal wurtzite structure, exhibits excellent crystalline periodicity along its axial direction, forming an ideal structural foundation for the piezoelectric effect [277,278]. Under external force, positive and negative ions within the ZnO crystal undergo relative displacement, generating polarization and accumulating charges on the material surface, thereby producing a piezoelectric potential [266].

He et al. [279] developed piezoelectric smart composites by integrating PEDOT and copper thiocyanate (CuSCN)-coated ZnO nanorods onto carbon fiber (CF) surfaces (Figure 5d). In this composite, PEDOT served as a crucial conductive layer, forming a P-N junction with the n-type ZnO nanorods. This configuration not only facilitated the deposition of the gold electrode but also significantly enhanced piezoelectric performance. Following PDMS encapsulation, the composite operated as an energy harvester and self-powered sensor capable of detecting vibrations. The output voltage increased from 1.4 V to 7.6 V as acceleration rose from 0.1 m s^−2^ to 0.4 m s^−2^, demonstrating highly reliable performance even after 100,000 impact tests and 65,000 dynamic vibration tests.

Similarly, Li et al. [280] fabricated a three-layer heterojunction structure by sandwiching a PEDOT:PSS layer between n-type Si and ZnO nanowires. They utilized the piezo-phototronic effect—arising from positive and negative piezoelectric charges—to enhance the performance of Si/ZnO/PEDOT:PSS photodetectors. PEDOT:PSS formed strong interface contacts with ZnO and modulated the generation, separation, transport, and recombination of photogenerated carriers through piezoelectric charges. Under external compressive strain, the positive and negative piezoelectric charges generated by the ZnO nanowires acted on the n-Si/n-ZnO and n-ZnO/p-PEDOT:PSS interfaces, respectively, enhancing the photoresponse performance by over 30-fold. This study provided critical experimental and theoretical insights into the application of piezo-phototronic effects—particularly the coupling of positive and negative piezoelectric charges—in multilayer devices.

**Figure 5 molecules-30-00179-f005:**
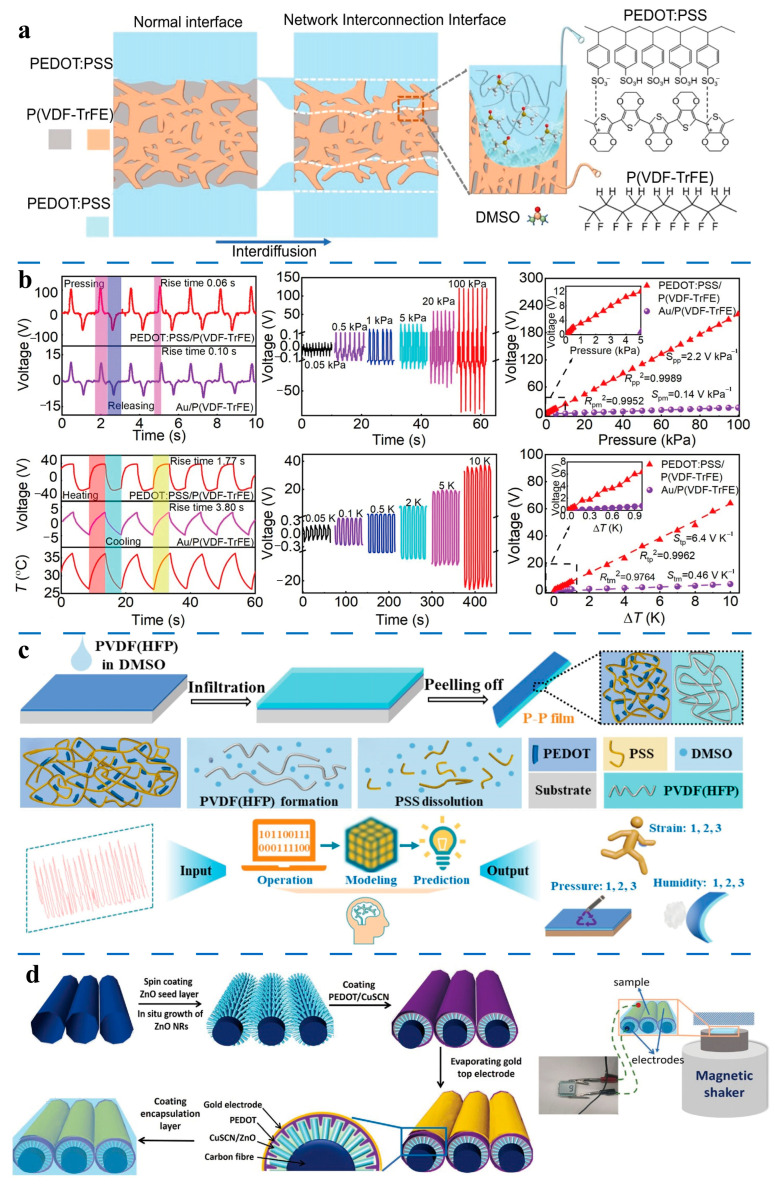
(**a**) Schematic illustration of the formation of the network interconnection interface in the P(VDF-TrFE)/PEDOT:PSS film. (**b**) Piezoelectric and pyroelectric voltage output of PEDOT:PSS/P(VDF-TrFE) with the NII [275]. (**c**) Schematic illustrations of the fabrication process of the PEDOT:PSS-PVDF(HFP) film and their uses as strain, pressure, and humidity sensors combined with machine learning for multimodal identification [276]. (**d**) Fabrication process of PEDOT/CuSCN/ZnO CF nanogenerator and the scheme of harvesting energy from impacting acceleration of 0.1 m s^–2^ to activate LCD screen [279].

### 3.2. Electrochemical Properties

Beyond their electrical properties, PEDOTs also exhibit remarkable electrochemical activity and stability, which make them highly promising for long-term applications in electrochemical devices such as batteries, ECDs, supercapacitors, and more [281,282,283,284]. For example, Charkhesht et al. [285] demonstrated that PEDOT:PSS can function as a conductive binder in lithium (Li)-ion batteries, significantly enhancing the electrochemical performance of metal electrodes. The study showed a high electrochemical capacity with a gravimetric capacity of approximately 302 mAh g^−1^ at 0.2 C, maintaining stable cycling performance even after 100 cycles.

The redox behaviors of PEDOTs play a crucial role in their electrochemical performance. The transition between the doped (conductive) and dedoped (non-conductive) states involves the gain or loss of electrons or the redistribution of shared electron pairs. Doping significantly alters the energy band structure and charge transport properties of PEDOTs. In the doped state, PEDOTs accept foreign electrons or ions (e.g., anions), which transform into charge carriers. This process modifies the electronic structure of the molecular chains, forming charge transport pathways and enhancing electrochemical activity along with electrical conductivity. Conversely, dedoping leads to the loss of these electrons or ions, decreasing conductivity but potentially improving electrochemical activity. Thus, the redox reaction rate—not just charge concentration—reflects the electrochemical activity of PEDOTs. A faster redox reaction indicates improved responsiveness to external electric stimuli, enhancing the performance of electrochemical devices.

Several factors influence redox behavior, including the purity, crystallinity of PEDOTs, and the type and concentration of the electrolyte. Adjusting the manufacturing process, increasing material crystallinity, or selecting suitable counter-anions can optimize these properties. PEDOTs also exhibit high redox reversibility, enabling repeated switching between oxidized and reduced states without significant performance degradation. This characteristic is critical for the long-term stability of electrochemical devices.

Functionalizing PEDOT with antifouling groups, such as tri(ethylene glycol) (EG3), tetra(ethylene glycol) (EG4), sulfobetaine (SB), or phosphorylcholine (PC), allows for tailored electrochemical properties (Figure 6a) [286]. In electrochemical impedance spectroscopy (EIS) measurements over a frequency range of 1 Hz to 100 kHz, all PEDOT-based electrodes (PEDOT, PEDOT-EG3, PEDOT-EG4, PEDOT-SB, PEDOT-PC) demonstrated lower impedance compared to gold (Au) electrodes (Figure 6b). This indicates that PEDOT and its derivatives outperform metal Au in minimizing energy loss during signal recording and transmission. Functionalization with EG3 and PC groups maintained the inherently low impedance of PEDOT, resulting in superior electrochemical stability and small voltage deviations under electrical stimulation.

The electrochemical properties of PEDOT are closely tied to the oxidation level and conjugation length of its molecular chains. For instance, Chen et al. [287] explored the effects of different oxidants (FeCl_3_, Fe(Tos)_3_, and MoCl_5_) on PEDOT’s chain structures and capacitive performance during the vapor-phase polymerization of EDOT (Figure 6c). Fe(Tos)_3_ notably increased PEDOT’s oxidation degree and conjugation length, enhancing electrical conductivity and creating more active polymerization sites. Electrodes prepared with Fe(Tos)_3_ exhibited a conductivity of 73 S cm^−1^, a high areal capacitance of 419 mF cm^−2^, and excellent electrochemical stability under various bending conditions (Figure 6d). Moreover, the extended conjugated structures strengthened inter-chain interactions, contributing to improved cycling stability.

Supercapacitors, also known as electrochemical capacitors, store charge through the formation of an electric double layer or by enabling redox reactions near the electrodes [288]. For PEDOTs, charge storage occurs primarily through redox reactions, with performance evaluated based on parameters such as specific capacitance, energy density, and power density. PEDOTs exhibit excellent electrical conductivity and environmental stability, which contribute to enhanced charge storage efficiency and long-term cycling stability. In contrast, carbon nanomaterials, such as graphene-based supercapacitors, typically offer ultra-high specific capacitance and superior rate performance, enabling rapid charging and discharging over short durations [289]. However, graphene’s high preparation costs and the tendency of its nanosheets to aggregate and disperse unevenly result in a reduction in effective specific surface area and significant decreases in charge storage capacity. Metal oxides, such as MnO_2_, also play a crucial role in electrochemical energy storage due to their abundant redox-active sites, which provide high specific capacitance and energy density [290]. However, MnO_2_ can undergo structural changes during cycling, leading to decreased cycling stability and limiting its suitability for long-term cycling applications.

Combining PEDOT with other conductive polymers or inorganic nanomaterials forms composites that further enhance charge storage capacity and cycling stability. These composites capitalize on the complementary properties of each component to achieve optimized performance. For example, in PEDOT/graphene composites, PEDOT prevents graphene aggregation while graphene offers conductive pathways and a large specific surface area for PEDOT, synergistically enhancing charge storage capacity. This makes the hybrid material superior to single PEDOT-based systems in terms of charge storage performance [291]. Furthermore, graphene’s high stability provides robust structural support for the hybrid material, mitigating structural changes in PEDOT during cycling and significantly improving overall cycling stability. In one study, Yeon et al. [292] treated PEDOT:PSS films with nitric acid (HNO_3_) at a concentration of 14 mol L^−1^ for 10 min at room temperature, achieving an exceptional electrical conductivity of 4100 S cm^−1^. This conductivity made the material suitable as a counter electrode in Pt/fluorine-doped tin oxide (FTO)-free dye-sensitized solar cells (DSSCs, Figure 6e). The high conductivity facilitated efficient charge transport within the electrode, enhancing capacitive performance (Figure 6f) and achieving a PCE of 8.59%. This highlights the potential of PEDOT:PSS as a low-cost, high-performance, and environmentally friendly alternative for DSSCs.

PEDOTs exhibit excellent stability during electrochemical processes, maintaining their electrical conductivity in the oxidized state over extended periods, making them ideal for high-performance supercapacitor electrodes [293]. By employing specific preparation techniques, such as compositing with Au nanoparticles [294], PEDOT and its derivatives can achieve superior charge/discharge performance. For instance, their discharge-specific capacitance remains high even at increased current densities, enabling rapid charging and discharging (Figure 6g,h) [295,296]. Moreover, the capacitance value of PEDOT electrodes shows minimal degradation after numerous charge/discharge cycles, ensuring the long-term cycling stability of supercapacitors [297].

Additionally, PEDOTs typically feature a wide voltage window across different electrolyte systems, allowing supercapacitor electrodes to operate within a large voltage range, thereby improving energy density [298]. The hole density in PEDOTs, particularly PEDOT:PSS, can be reversibly modulated over a wide range by ion compensation provided by the electrolyte. This makes PEDOTs highly effective as electrodes in ion-based energy storage devices, including supercapacitors, and as channels in electrochemical transistors and sensors [288]. The reversible modulation of hole density is also accompanied by a color change from deep blue to transparent, enabling PEDOT:PSS to function as a pixel electrode in ECDs [299]. PEDOTs also possess excellent processability and flexibility, enabling the fabrication of electrode materials in various shapes and sizes. These properties make PEDOT-based electrodes ideal for designing and manufacturing wearable, miniaturized supercapacitors [300]. PEDOTs exhibit excellent cellular and tissue compatibility. They not only establish effective contact and interaction with biological tissues but also retain their outstanding electrical properties, which strongly support the transmission and processing of biological signals. This makes them a key material in the field of electrochemical sensing and detection [78,301]. In biomedical research, PEDOT:PSS has been extensively utilized to fabricate various in situ cellular electrochemical sensors for detecting active small molecules, such as dopamine and hydrogen peroxide, released by cells [302,303]. Furthermore, PEDOT:PSS has been employed in the development of wearable electrochemical sensors for real-time biomarker detection in sweat, urine, and other bodily fluids [304,305,306].

**Figure 6 molecules-30-00179-f006:**
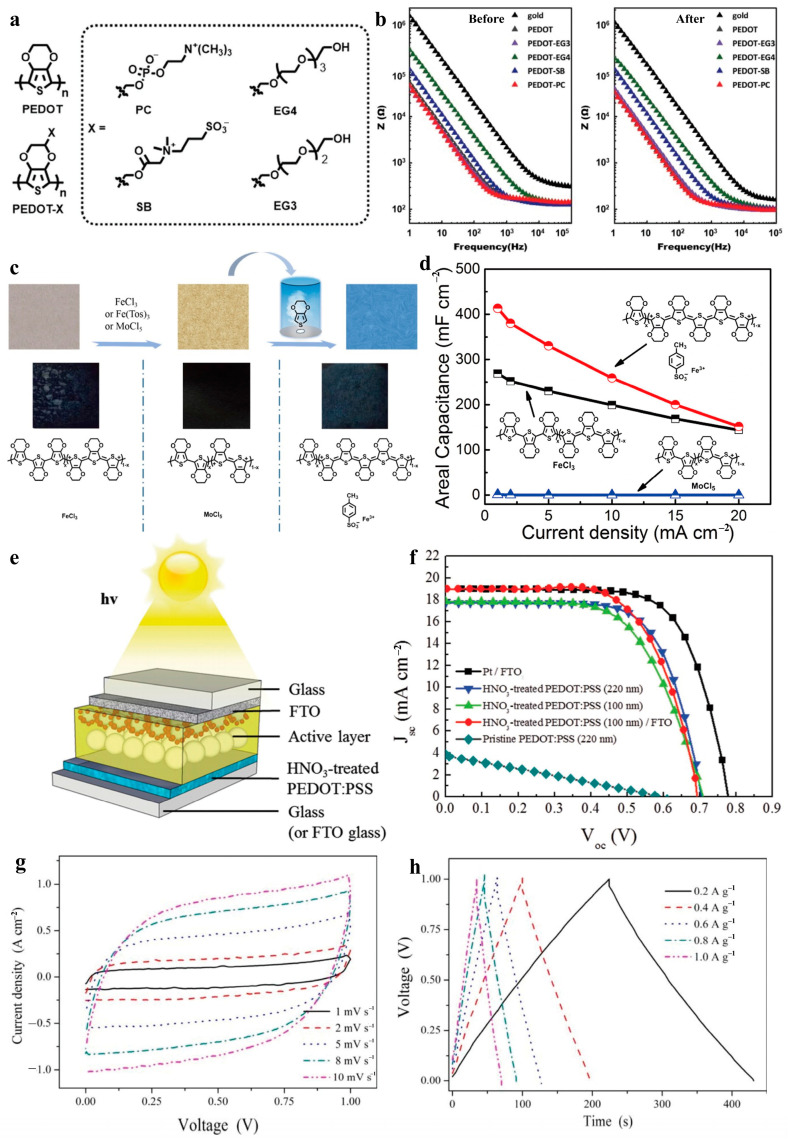
(**a**) Chemical structures of the antifouling PEDOTs. (**b**) EIS of the gold, PEDOT, PEDOT-EG3, PEDOT-EG4, PEDOT-SB, and PEDOT-PC electrodes before and after the biphasic stimulation [286]. (**c**) Schematic representation of the preparation of PEDOT/P(FeCl_3_), PEDOT/P(MoCl_5_), and PEDOT/P(Fe(Tos)_3_) via vapor-phase polymerization, as well as the structure change in PEDOT chains. (**d**) Comparison of the capacitance performance of EDOT monomer assemblies regulated by different oxidizing agents [287]. (**e**) Schematic structure of the DSSC incorporating the HNO_3_-treated PEDOT:PSS as an electrode. (**f**) The J-V characteristic curves of DSSCs with different electrodes: PEDOT:PSS-only, PEDOT:PSS/FTO, and Pt/FTO [292]. (**g**) Cyclic voltammetry curves of the PEDOT supercapacitor with different scanning rates. (**h**) Galvanostatic charge/discharge curves of the PEDOT supercapacitor at different current densities [295].

## 4. Magnetic Properties of PEDOTs and Structural Tunability

Magnetic materials, particularly nanomaterials, with properties such as superparamagnetism and magnetically induced quantum tunneling effects, have found extensive applications in various fields, including magnetic refrigeration, magnetic fluids, high-density information storage, color imaging, cell separation, medical diagnostics, targeted drug delivery, sensors, food or pharmaceutical testing, environmental purification, photocatalysis, and military aviation [307,308,309,310,311]. Currently, inorganic magnetic materials, particularly ferrites, dominate this space. Among them, Fe_3_O_4_ nanoparticles stand out due to their ease of synthesis, high coercivity, superparamagnetism, biocompatibility, and non-toxic nature [312]. In contrast, the magnetism of organic polymers falls under the domain of organic spintronics. Since the first report of a purely organic magnetic polymer—a polydiacetylene substituted with stable radicals—nearly four decades ago, hundreds of materials have been discovered, categorized primarily as structural or composite types [313]. However, most structural materials exhibit ferromagnetism only at low temperatures, and their magnetic sources differ from those of traditional inorganic materials, leaving theoretical understanding incomplete. Room-temperature ferromagnetic organic semiconductors remain a rarity, with few breakthroughs reported [314].

Composite materials involving CPs are gaining significant attention. External magnetic fields have been found to influence the synthesis of CP-based composites, such as PEDOT/polysulfone dianiline (PSDA), thereby affecting their electrical conductivity and Seebeck coefficient [315]. At temperatures below 30 K, doped intrinsically conducting polymers (ICPs) like PANi and certain copolymers exhibit magnetic properties, but these are not yet practical for applications [316,317]. Since the 1990s, research has shifted toward developing organic–inorganic magnetically conductive polymer composites (MCPCs). These materials endow CPs—especially PANi, PPy, PEDOT, and PEDOT:PSS—with magnetism by incorporating magnetic nanoparticles or doping with magnetic ions [318,319,320,321]. These composites, which may feature dual/multi-component or dual/multi-layer structures, combine CPs with nanoparticles such as metals (Ni, Co, Fe), metal salts (AgCl, LiFePO_4_), metal oxides (γ-Fe_2_O_3_, Fe_3_O_4_, M-type barium ferrite), ferrites, and carbon-based materials (e.g., carbon black, carbon nanotubes, graphene) [320,321,322].

MCPCs combine the optoelectronic properties of CPs with the magnetic properties of inorganic magnetic materials. Additionally, they integrate the unique advantages of nanomaterials and organic polymers. The CP coatings serve to prevent the aggregation of inorganic magnetic nanoparticles, which otherwise tend to cluster due to their high surface activity. Moreover, these coatings enhance the moldability and processability of such composites. As a result, MCPCs have emerged as a new generation of functional materials for the development of optoelectromagnetic systems [323,324,325]. Their excellent processing and film-forming properties, inherited from CPs, enable diverse applications, including electromagnetic shielding, radar absorption, stealth coatings, high-capacity information storage, low-loss high-frequency microwave communication, sensors, targeted drug delivery, electrochromism, nonlinear optics, molecular electronics, and nanomotors [325,326,327,328,329].

PEDOT-based MCPCs are typically formed by integrating various inorganic magnetic powders (such as ferrites and rare-earth alloys) and additional components (e.g., binders) into a PEDOT matrix. These composites exist in diverse forms, including films, fibers, polymer microspheres, and hydrogels [322,330,331,332]. Their electrical conductivity primarily arises from PEDOT, while their magnetic properties are attributed to the incorporated magnetic particles. The electrical properties of PEDOT-based MCPCs depend on numerous factors, such as the molecular backbone structure, type of dopant, doping level, morphology, and synthesis method of PEDOT [333]. Similarly, the type, size, morphology, binding mode with PEDOT, and content of inorganic particles critically influence the magnetic properties of the composite.

Typically, the electrical conductivity of MCPCs ranges from 10^−4^ to 10^2^ S cm^−1^. However, as the content of magnetic nanoparticles increases, carrier transport between polymer chains is hindered, leading to a decrease in electrical conductivity. In contrast, the saturation magnetization of the composite increases consistently with the nanoparticle content. This phenomenon also correlates with the demagnetization field effects caused by the non-magnetic CPs [334]. Most PEDOT-based MCPCs exhibit superparamagnetic behavior, although this is not universal and may vary with temperature. For instance, Pt_3_Co-PEDOT:PSS films exhibit electrical conductivities between 1.6 and 4.0 S cm^−1^. They demonstrate superparamagnetism above the blocking temperature (TB, 110.5 K) and transition to ferromagnetic behavior when the temperature falls below TB [334].

The adjustment of electromagnetic properties in PEDOT-based MCPCs largely depends on their preparation strategies. Beyond controlling structural parameters—such as the composition and content of magnetic nanoparticles, aggregation morphology, particle size distribution, interfacial interactions, and spatial uniformity among components—the design of special architectures like core–shell structures and coaxial nanowires plays a crucial role [335,336]. Currently, liquid-phase methods dominate the preparation techniques for PEDOT-based MCPCs. These methods include direct blending [337], in situ synthesis (e.g., in situ polymerization) [338], hard-template synthesis [339], self-assembly using soft templates [340], and electrospinning [341].

For instance, Naim et al. [342] developed CTe/PEDOT:PSS films via blending, embedding carbon dot-decorated tellurium nanorods into a PEDOT:PSS molecular network. The tellurium nanorods exhibited intrinsic electromagnetic properties, enhancing the absorption and scattering of electromagnetic waves. Meanwhile, the PEDOT:PSS matrix formed a conductive network that directed electromagnetic waves into the material, where they were absorbed and dissipated through internal interactions. Similarly, spin-coating an aqueous mixture of PEDOT:PSS and Fe_3_O_4_ nanoparticles produced highly conductive and superparamagnetic PEDOT:PSS/Fe_3_O_4_ films with excellent transparency, mechanical flexibility, and lightweight characteristics [337]. After treatment with methylammonium iodide (MAI)/DMF solution, these films demonstrated a saturation magnetization of 43 emu g^−1^ and electrical conductivity reaching 1080 S cm^−1^, making them effective at shielding electromagnetic radiation. Moreover, these films could be coated onto flexible silk and cotton threads to impart electromagnetic properties (Figure 7a).

However, blending, while being a straightforward and simple processing method, poses challenges in controlling the aggregation and dispersion uniformity of magnetic nanoparticles. This often results in structural and performance uncertainties within the composites. To address these issues, auxiliary dispersion techniques such as ultrasonic processing are frequently employed. Additionally, surfactants and coupling agents are used to modify the surface of nanoparticles, enhancing their stability and dispersion. For example, blending polyvinyl butyral (PVB) with PEDOT:PSS-Gd_5_Si_4_ improved the composite system’s stability and expanded the microwave absorption bandwidth of the coating film [343].

The in situ polymerization method offers a straightforward solution for fabricating composites, particularly effective in reducing nanoparticle aggregation and promoting the formation of core–shell structures. This approach also enhances the thermal stability and mechanical properties of the resulting materials. For instance, Feng et al. [338] developed electromagnetic lignocellulosic composites by in situ polymerizing a PEDOT coating alongside magnetic Fe_3_O_4_ particles within delignified wood (Figure 7b). The three-dimensional (3D) porous structure of this composite effectively attenuates electromagnetic waves through multiple internal reflections, aided by the synergistic effects of magnetic loss from Fe_3_O_4_ particles and dielectric loss from the PEDOT layer. The resulting material exhibited outstanding properties, including excellent electrical conductivity (10^3^ S m^−1^), significant magnetism (26.7 emu g^−1^), and an electromagnetic shielding (EMS) effectiveness of up to 59.8 dB. Similarly, Taj et al. [344] synthesized composites of PEDOT and graphene nanosheets via in situ chemical oxidative polymerization. Post-doping, the chemical bonding and enhanced interactions between PEDOT and graphene enabled superior impedance matching across a broad frequency range. This facilitated the entry of electromagnetic waves into the composite, where they were either absorbed or reflected. Furthermore, as the frequency increased, the EMS effectiveness of the composite steadily improved.

**Figure 7 molecules-30-00179-f007:**
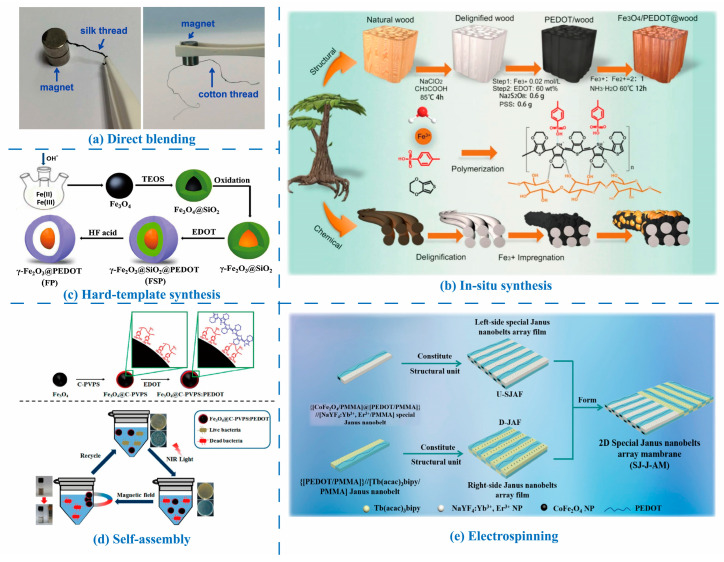
(**a**) Mechanical response of silk and cotton threads coated with PEDOT:PSS/Fe_3_O_4_ prepared by direct blending method [337]. (**b**) Schematic diagram of in situ polymerization of Fe_3_O_4_/PEDOT@wood [338]. (**c**) Schematic illustration of the preparation of hollow γ-Fe_2_O_3_@PEDOT nanocomposites via hard template method [339]. (**d**) Schematic diagram of the self-assembly synthesis of Fe_3_O_4_@C-PVPS:PEDOT, and schematic representation of NIR irradiated antibacterial activity followed by application of a magnetic field to isolate and recycle Fe_3_O_4_@C-PVPS:PEDOT [340]. (**e**) Schematic diagram showing electrospinning preparation of PEDOT-based special Janus array membrane (SJ-J-AM) [341].

Unlike conventional film-forming techniques, the hard-template method has been employed to construct PEDOT-based MCPCs with unique spherical morphologies. For example, hollow γ-Fe_2_O_3_@PEDOT core–shell nanospheres with significantly enhanced microwave absorption properties were fabricated using γ-Fe_2_O_3_@SiO_2_@PEDOT core–shell nanocomposites as templates, which were reacted with hydrofluoric acid (HF) (Figure 7c) [339]. The hollow structure provided several benefits, including a larger specific surface area, reduced density, and the ability to accommodate more Fe_2_O_3_ within the internal cavity. Despite these advantages, the hard-template method has its drawbacks, such as the complexity of template removal and the risk of damaging the target product’s morphology due to polymeric nanostructure aggregation during the process [345].

In contrast, the self-assembly approach can be seen as a soft-template method, utilizing non-covalent interactions such as hydrogen bonding, van der Waals forces, ionic bonds, and coordination bonds to spontaneously assemble molecules, atoms, or nanoparticles into stable, ordered structures within a specific chemical environment [309]. Potential soft templates include micelles, liquid crystals, polyelectrolytes, and colloids. This method offers advantages such as simplicity, uniform composition, controllable structures with minimal defects, and the ability to mimic biomembranes, making it a cutting-edge technique in the development of PEDOT-based MCPCs [346].

For instance, Fe_3_O_4_ nanoparticles were first coated with catechol-conjugated poly(vinylpyrrolidone) sulfobetaines (C-PVPS) to impart negative surface charges, followed by encapsulation with PEDOT via a layer-by-layer (LBL) self-assembly process (Figure 7d) [340]. The resulting Fe_3_O_4_@C-PVPS:PEDOT recyclable photothermal nanoparticles exhibited high stability and recyclability, allowing for rapid separation and reuse using an external magnetic field. These nanoparticles also demonstrated excellent near-infrared (NIR) photothermal conversion efficiency, effectively killing 99% of *Staphylococcus aureus* and *Escherichia coli* within just 5 min.

Electrospinning technology enables the preparation of micron- and nanoscale fibers of PEDOT-based MCPCs with oriented alignment and uniform distribution on substrates by applying a high-voltage electric field [308]. Factors such as polymer viscosity and molecular weight significantly influence the spinning process [347]. For instance, using sequential coaxial and side-by-side electrospinning techniques, PEDOT-based dual anisotropic Janus nanobelts (SJ-J-AM) were fabricated, featuring up/down-conversion fluorescence and electromagnetic properties (Figure 7e) [341]. This concentric uniaxial partitioned structure efficiently separates PEDOT, magnetic nanoparticles (CoFe_2_O_4_), and luminescent materials (e.g., NaYF_4_:Yb^3+^, Er^3+^, and Tb(acac)_3_bipy), minimizing adverse interactions among the composite components. This design enhances fluorescence efficiency and improves electrical conductivity.

Electrospinning technology can be utilized to prepare micron/nanoscale fibers of PEDOT-based MCPCs on substrates with oriented alignment and uniform distribution, using a high-voltage electric field [308]. Conditions such as polymer viscosity and relative molecular mass have a significant impact on the spinning process [347]. For example, through a sequential coaxial and side-by-side electrospinning technique, a PEDOT-based dual anisotropic special Janus array membrane (SJ-J-AM) with both up/down-conversion fluorescence and electromagnetic properties was prepared (Figure 7e) [341]. This concentric uniaxial partitioned structure effectively can separate PEDOT, magnetic nanoparticles (CoFe_2_O_4_), and luminescent materials (such as NaYF_4_:Yb^3+^, Er^3+^, and Tb(acac)_3_bipy), reducing adverse interactions among the composite systems and enhancing fluorescence efficiency and electrical conductivity.

There are additional solid-phase methods, such as high-energy ball milling [348], as well as gas-phase approaches like oxidative molecular layer deposition (oMLD) [349] and molecular-atomic deposition (MAD) [350]. Emerging specialized techniques, such as microwave-assisted surface imprinting (MSI) [351], are also employed to construct PEDOT-based MCPCs. For instance, oMLD enables the production of 1D Fe_3_O_4_-PEDOT nanospindles (Figure 8a) [349]. By adjusting the number of PEDOT cycles, the shell thickness can be precisely controlled. The spindle architecture, with its core–shell structure, facilitates increased microwave internal reflections, scattering, and conductive network formation, ultimately resulting in significantly enhanced microwave absorption intensity.

**Figure 8 molecules-30-00179-f008:**
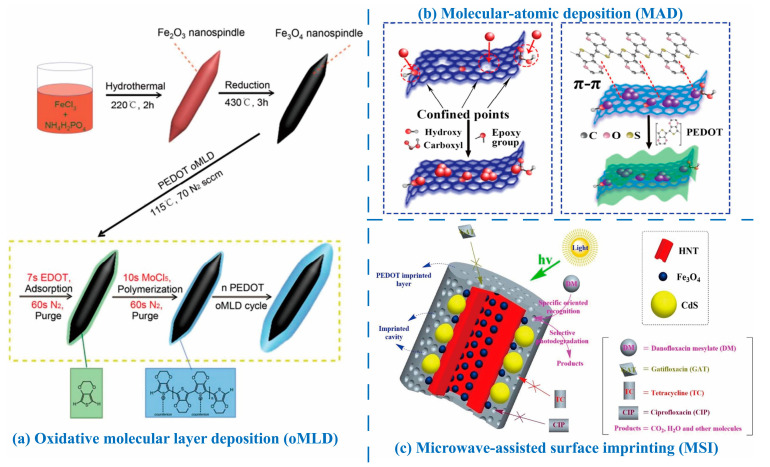
(**a**) Schematic diagram of oxidized molecular layer deposition for the preparation of Fe_3_O_4_-xPEDOT nanospindles [349]. (**b**) The illustration of atomic and molecular deposition mechanisms [350]. (**c**) Magnetically imprinted PEDOT/CdS heterojunction photocatalytic nanoreactors for three-dimensional specific recognition of danofloxacin mesylate (DM) [351].

Using the MAD technique (Figure 8b), the electromagnetic properties of 2D PEDOT:PSS-Fe_3_O_4_-rGO powder can be finely tuned, and their microwave absorption capabilities enhanced by optimizing the deposition and hybridization ratio of atoms and molecules [350]. This approach achieved a 140% expansion in effective absorption bandwidth, with a maximum reflection loss of –61.4 dB. These improvements are attributed to PEDOT:PSS deposition, which reconstructs the polymer-induced charge transport network to provide excellent dielectric loss, as well as the multiple relaxation effects introduced by the hybridized interfaces.

The MSI method facilitates the production of PEDOT/CdS p-n heterojunction nanoreactors with a well-defined hollow tubular structure, strong adsorption capacity, efficient light absorption, excellent photocatalytic activity, and recyclability (Figure 8c) [351]. These nanoreactors enable 3D-specific recognition and selective photocatalytic degradation of danofloxacin mesylate (DM). The PEDOT-imprinted layer not only absorbs photoinduced holes generated by CdS but also encapsulates the structure, effectively preventing photocorrosion and secondary pollution. Additionally, the introduction of Fe_3_O_4_ into PEDOT/CdS enhances the transfer of photoexcited electrons, increasing the degradation rate from 77.16% to 84.84%.

In summary, PEDOT-based MCPCs exhibit multifunctionality across optics, electronics, magnetics, and nanotechnology, positioning them as one of the most compelling and challenging research fields with extensive application potential. Current research primarily focuses on composites of metal microparticles and ferrites with PEDOT or PEDOT:PSS, utilizing diverse fabrication methods (Figure 9). Beyond optimizing composition and microstructure, the development of multivariate and multilayered composites presents significant opportunities for future applications. Additionally, the exploration of electromagnetic materials derived from pure CPs remains a promising area for continued investigation.

**Figure 9 molecules-30-00179-f009:**
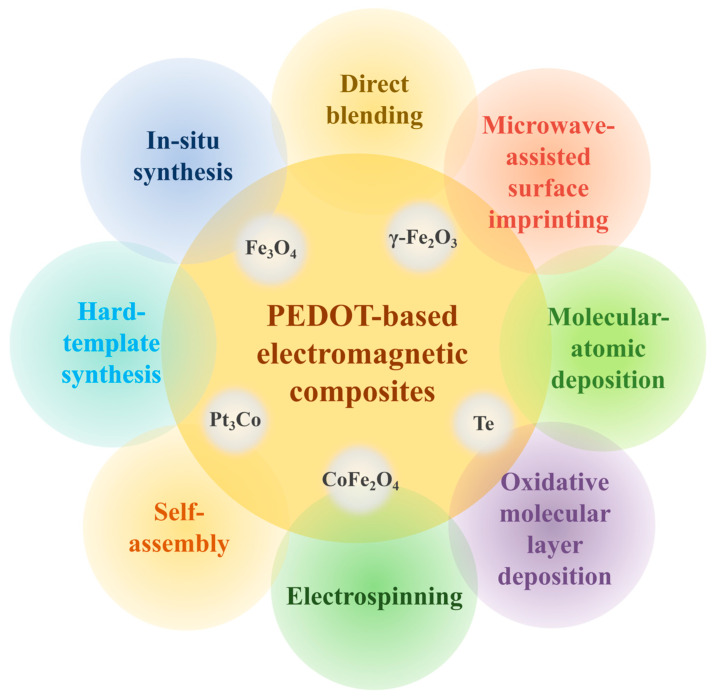
Methods for preparing PEDOT-based electromagnetic composites.

## 5. Conclusions and Perspective

This article provides a comprehensive overview of the optoelectromagnetic properties of PEDOTs, detailing the underlying mechanisms and tuning strategies for their diverse applications. PEDOTs demonstrate exceptional optical transparency, light absorption, and emission characteristics, making them highly suitable for organic optoelectronic devices such as antistatic coatings, transparent electrodes, OSCs, OLEDs, and ECDs. PEDOTs, particularly PEDOT:PSS, exhibit excellent electrical and thermoelectric performance, which can be enhanced through various strategies, including doping (or compositing) with polar solvents, ionic liquids, or surfactants, and post-treatment techniques involving light, heat, acids, bases, or solvents. These properties make them indispensable in organic electronics, especially in the development of highly conductive films and energy conversion devices.

Although PEDOT lacks intrinsic piezoelectricity due to its crystal structure, piezoelectric effects can be introduced via dopant modification or compositing, enabling applications in piezoelectric sensors and energy harvesters. Their remarkable electrochemical activity and stability also make PEDOTs ideal for applications in supercapacitors, sensors, and OSCs. In the electromagnetic domain, PEDOT composites with metal powders, ferrites, and carbon materials have been widely studied, finding use in electromagnetic shielding, microwave absorption, and infrared stealth technologies.

The interplay between optoelectromagnetic properties and factors such as mechanical characteristics (e.g., flexibility, stretchability, elasticity) and surface–interface properties (e.g., wettability, roughness, adhesivity, densification) presents both opportunities and challenges. The synergistic integration of optoelectromagnetic functionality and mechanical properties is increasingly critical for enhancing the environmental stability, operational reliability, and multifunctionality of PEDOT-based materials in diverse applications, particularly in flexible and wearable electronics.

Mechanical properties can be further enhanced by optimizing interactions between molecular chains or interfaces involving PEDOT and conductive fillers, polymer matrices, or other components. In addition to refining preparation processes, novel structural strategies—such as multilayer construction, heterojunctions, crosslinking networks, self-assembly, advanced mixing, and printing techniques—offer promising avenues for improvement. These strategies must be carefully tailored to meet specific application requirements. For instance, flexible photoelectric sensors must combine optical properties for signal detection, electrical properties for efficient signal transmission, and mechanical robustness for adaptability to diverse environmental and deformation conditions. Achieving this combination will make PEDOT-based materials highly suitable for applications like intelligent robotics (e.g., haptic perception) and biomedical imaging (e.g., flexible detectors). Integrating these multifunctional properties into simplified device structures could significantly enhance performance while maintaining compactness and efficiency.

Addressing compatibility issues in composite systems remains crucial to preserving PEDOTs’ excellent optoelectromagnetic properties. Furthermore, PEDOTs’ versatility—manifesting in forms such as films, fibers, and gels—will drive innovations in wearable, flexible, stretchable, and biocompatible electronics. The future development of PEDOT-based materials hinges on the synergistic optimization of their optoelectromagnetic, mechanical, surface–interface, and other properties. This will require addressing current technological challenges and fostering interdisciplinary research to explore novel material architectures and processing techniques.

PEDOTs’ ability to bridge traditional boundaries between photonics, electronics, magnetics, and mechanics positions them as a cornerstone for the next generation of multifunctional materials. By enabling seamless integration into systems with complex demands for sensing, energy harvesting, and actuation, PEDOTs have the potential to revolutionize fields such as autonomous and soft robotics, personalized healthcare, and advanced prosthetics. Advancements in PEDOT research will not only address current application challenges but also pave the way for transformative innovations across a wide spectrum of emerging technologies.

Looking ahead, further exploration of material composition and microstructure, along with the design of multi-dimensional and multi-layer composites, will enhance application value. Innovations in processing technologies—such as additive manufacturing (3D/4D printing), photolithography, and surface plasma treatment—will be pivotal. Deeper investigations into the mechanisms governing optoelectromagnetic properties, supported by theoretical calculations and experimental research, will also be essential. Advanced tools like generative artificial intelligence, machine learning, and neural networks can accelerate breakthroughs, enabling structure-property-function modeling to optimize PEDOTs’ design and performance.

Future research should also address the long-term material stability and operational reliability of PEDOTs under environmental stressors, including temperature, humidity, dirt, corrosive media, oxygen, and UV exposure. Beyond encapsulation, the integration of biomimetic functionalities—such as self-healing, superhydrophobicity, and adaptability—will enhance durability and expand application scenarios. Smart PEDOTs with responsive properties, such as temperature-sensitive conductivity or light- and magneto-controlled behavior, hold potential for innovations in reversible switching devices, optical storage, and bioelectronic medical applications.

In conclusion, PEDOTs, with their unique optoelectromagnetic properties, are poised to play an increasingly significant role in organic electronics, driving innovations in fields ranging from wearable devices to smart, self-powered technologies. The continued exploration of their capabilities promises exciting advancements and new application directions.

## Data Availability

Not applicable.

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
