# Peer review of "Recent Advances in the Tunable Optoelectromagnetic Properties of PEDOTs"

_molecules, 2025, doi:10.3390/molecules30010179_

Round 1
Reviewer 1 Report
Comments and Suggestions for Authors
Comments
In this work, Zhu and colleagues provide a comprehensive overview of the development history of PEDOT, its tunable properties in optoelectromagnetic applications, and its broad potential for future applications, which could offer guidance for researchers in the field. In my opinion, this will be of interest to the audience of Molecules journal. However, there are some issues that need to be addressed.
1. The current title is not sufficiently clear, as 'tunability' can imply the ability to modify both optical and electromagnetic properties. To better reflect the overall concept and emphasize clarity without redundancy, a more precise formulation is needed, e.g., Recent Advances in the Tunability of Optoelectromagnetic Properties of PEDOTs.
2. As a scientific review paper, maintaining coherent and precise language is essential. It is recommended that the authors carefully revise and polish the language throughout the manuscript. For example, in the abstract, using "Electrically" and "Magnetically" alone at the beginning of sentences may feel incomplete or awkward. Rephrasing these parts would improve the overall flow and clarity of the text. Here's a few alternative examples: ‘In terms of optics, PEDOTs are..’, ‘From an electrical perspective’….
3. Some sections of the review article should focus more on proposing the underlying mechanisms rather than relying solely on references. For instance, in the summary of Section 2.5, the current discussions make it challenging to discern the correlations related to scattering properties.
4. The authors mention "biocompatibility" in line 58; however, the subsequent descriptions do not appear to elaborate on or connect with this aspect; The phrase " papers" in line 71 may sound a bit informal. It would be more appropriate to replace it with terms like " works, or studies", etc.; the term "paper" on line 99 may not be appropriate….as well as the font issue on line 973 that should be addressed.
5. Some sentences are suggested to be refined to improve clarity. For example, the first sentence of Section 2 could be modified to "PEDOTs are highly efficient in absorbing photons and generating charge carriers, with the majority of their applications utilizing them in film form.", Please review and revise the entire content accordingly.
6. Some section titles are suggested to be reviewed for readability and clarity. Here's an example for 3.1.1 'Modification Methods to Enhance Electrical Conductivity
7. Figure 3 is not accurate and should be revised. Additionally, the discussion section should integrate the analysis of the figure, rather than just briefly mentioning it.
8. It may also be important to mention the mechanical properties, such as flexible/stretchable devices based on PEDOT. Moreover, PEDOT has been widely applied in the rapidly developing perovskite devices, as demonstrated in the related literature (https://doi.org/10.1002/EXP.20220006; https://doi.org/10.1117/1.APN.2.4.044001; https://doi.org/10.1117/1.AP.5.1.016001). These aspects could also be worth considering in the perspective section.
Reviewer 2 Report
Comments and Suggestions for Authors
The article examines recent advancements in the optoelectromagnetic properties of PEDOT and its derivatives, emphasizing their applications in organic solar cells, electrochromic devices, advanced sensors, and flexible technologies. It discusses strategies to enhance the performance of PEDOT:PSS, such as doping, chemical and physical treatments, and the development of hybrid composites with nanomaterials. The potential of PEDOT in flexible and portable electronic devices is highlighted, along with the need to optimize the balance of its electrical, optical, and mechanical properties to maximize its applicability in emerging technologies.
Comments
The review presented by the authors is comprehensive and insightful, effectively highlighting key properties of PEDOT:PSS. However, certain aspects could benefit from a more detailed discussion to enhance the analysis.
Section 2.1 “Optical properties and structural tunability of PEDOTs”
A notable characteristic of PSS in PEDOT:PSS is its acidity, which can corrode metals such as aluminum and compromise the stability of organic materials in multilayer devices like OLEDs and OSCs. This can lead to premature failures in devices where the interaction between layers is critical to ensuring performance and durability. In the case of organic solar cells, the acidity of PSS can degrade donor or acceptor layers, making it necessary to implement blocking layers or chemical modifications to mitigate this effect. It would be beneficial for the authors to address this aspect in their discussion.
Section 2.2 “Spectral transparency property”
Although PEDOT:PSS can achieve conductivities of up to 1000 S/cm through advanced modifications, this value remains significantly lower than that of indium tin oxide (ITO), which exceeds 10,000 S/cm. This limits its application in scenarios requiring ultrathin, highly conductive layers and often necessitates additional techniques, such as removing excess PSS. Furthermore, the durability of PEDOT:PSS is lower compared to ITO, as it tends to degrade over time under adverse conditions, such as high humidity or prolonged exposure to ultraviolet light. It would be valuable for the authors to delve deeper into this aspect, as while PEDOT can serve as a flexible substrate, it cannot always fully replace ITO.
Section 3.1.1 “Modification Methods to Enhance Electrical conductivity”
The non-crosslinked structure of PEDOT makes it prone to cracks and defects under repeated mechanical stress, limiting its use in flexible or portable devices requiring mechanical durability.
Although the doping of PEDOT is discussed, the inherent limitations or challenges associated with the process are not addressed in sufficient detail.
Section 3.1.3 “Piezoelectricity and Performance Enhancement”
While interfacial configurations such as the "network interconnection interface (NII)" and heterojunctions are highlighted, the text lacks detailed discussion on how these interfaces might be affected by external factors such as temperature, humidity, or mechanical wear. Additionally, a comparison with other structures would provide valuable context.
The authors should further elaborate and discuss how PEDOT and its specific compounds compare to other piezoelectric materials. Additionally, it would be valuable to address current commercial applications where this material is preferred over more traditional alternatives.
Section 3.2 “Electrochemical properties”
It would be beneficial to include in the text a comparison of PEDOT's properties, such as charge storage capacity and cyclic stability, with more advanced materials like graphene, metal oxides (e.g., MnO₂), or hybrid materials.

Round 2
Reviewer 1 Report
Comments and Suggestions for Authors
Before the manuscript proceeds further in the review process, several issues should be addressed:
- While the reference section includes numerous studies on perovskite solar cells, the main text only mentions this topic once. This inconsistency is neither scientifically rigorous nor reasonable.
- The claim to provide "an elaboration on the integration of optoelectromagnetic and mechanical properties" could be significantly strengthened by adopting a more forward-looking perspective that extends beyond illustrative examples.
- As far as we know, PEDOT has been widely utilized in optoelectronic devices, such as organic/inorganic light-emitting devices and promising printable technologies, which are undergoing rapid development. Given the close relevance of these applications to the subject of this review, it is unclear why this aspect has been overlooked. As mentioned in my earlier comments, I recommend including additional analysis or a concise summary of this topic.
- Although the manuscript was prepared with the assistance of a native English speaker, further refinement of the Abstract and Introduction would greatly enhance clarity and overall readability.
